# Targeted next-generation sequencing for comprehensive diagnosis and drug resistance detection in pulmonary and extrapulmonary tuberculosis: a single-center retrospective study

Jingyi Dai,[1,2] Qiujing Li,[1] Juan Wan,[1] Fengshuo Sun,[3,4] Gang Zhou,[3] Meiqiu Yang,[1] Chuanshu Dong,[1] Yao Fang,[1] Lifeng Li,[3] Lin Wang,[5] Guiming Liu[1]

**ABSTRACT**  Tuberculosis (TB), including drug-resistant forms, is a significant global health issue requiring accurate and rapid diagnostic tools. Traditional diagnostic methods suffer from low sensitivity and slow results, while nucleic acid amplification tests (NAATs) like Xpert MTB/RIF provide faster but incomplete solutions. Targeted next-generation sequencing (tNGS) has the potential to simultaneously detect *Mycobacterium tuberculosis*, drug resistance mutations, and co-infecting pathogens. This study evaluates the effectiveness of tNGS for TB detection across various pulmonary and extrapulmonary TB samples. We retrospectively enrolled 159 patients with suspected TB at Kunming Third People's Hospital. Specimens included 126 pulmonary and 33 extrapulmonary samples. All samples underwent tNGS, and results were compared against conventional microbiological tests and NAATs. tNGS demonstrated high diagnostic sensitivity for both pulmonary TB and extrapulmonary TB, achieving 83.9% sensitivity in bronchoalveolar lavage fluid, 89.5% in sputum, and 100% in extrapulmonary samples. tNGS showed an 83.3% agreement with Xpert in detecting rifampicin resistance. Additionally, tNGS also identified 22 drug resistance mutations, which are critical for predicting multidrug-resistant TB and pre-extensively drug-resistant TB. Additionally, tNGS effectively detected co-infecting respiratory pathogens, enhancing the understanding of complex TB cases. tNGS offers a highly sensitive and comprehensive approach for detecting TB and drug resistance, outperforming traditional methods in both pulmonary and extrapulmonary samples. It effectively identifies co-infections, providing a holistic view that enhances patient management, particularly in cases involving multidrug-resistant strains. These findings underscore tNGS's potential to improve TB diagnostics and patient management through faster and more precise detection methods.

**IMPORTANCE** Tuberculosis (TB) continues to be a leading cause of morbidity and mortality worldwide, particularly among marginalized populations. Timely and accurate diagnosis of TB, particularly extrapulmonary TB, remains challenging in low-resource settings, primarily due to non-specific clinical presentations that hinder early suspicion, along with limitations in both diagnosis and drug resistance detection. Additionally, targeted next-generation sequencing can identify co-infections with other clinically relevant pathogens, providing a more comprehensive understanding of each patient's infectious profile. By leveraging advanced sequencing technologies, our findings highlight a powerful diagnostic approach that can improve TB diagnosis, support appropriate treatment strategies, and may increasingly benefit patients in underserved settings as sequencing platforms become more accessible and costs continue to decrease.

Address correspondence to Lin Wang, 626182669@qq.com, or Guiming Liu, liuguimingkm@163.com.

Jingyi Dai and Qiujing Li contributed equally to this article. Author order was determined by alphabetically and in order of increasing seniority.

L.L., G.Z., and F.S. are employed by Genskey Inc., Beijing, China. The other authors declare that they have no conflict of interest to report.

See the funding table on p. 10.

KEYWORDS tuberculosis, drug resistance, targeted next-generation sequencing, pulmonary and extrapulmonary samples, diagnostic performance

Tuberculosis (TB), which mainly affects the lungs (pulmonary TB, PTB), can also occur in other parts of the body (extrapulmonary TB, EPTB), leading to substantial morbidity and mortality worldwide. In 2023, it was estimated that 10.8 million people developed TB, with 400,000 cases involving drug-resistant TB (DR-TB) (1). Early and precise identification of TB and DR-TB is crucial for effective treatment and for reducing the overall health burden. Delays in diagnosis contribute to worsening illness and increased mortality rates (2).

Traditional diagnostic methods like acid-fast bacilli (AFB) smear microscopy and mycobacterial culture are hindered by low sensitivity and prolonged result times. Molecular tests such as Xpert MTB/RIF allow for the rapid detection of *Mycobacterium tuberculosis* and rifampicin resistance but fall short in detecting a wider range of resistance mutations, which limits their effectiveness in managing DR-TB comprehensively (3).

Targeted next-generation sequencing (tNGS) focuses on specific genomic regions, providing a targeted approach to pathogen detection and resistance mutation identification. Its design reduces interference from background DNA, making it highly suitable for different types of samples. Metagenomic next-generation sequencing (mNGS) has been applied for the unbiased detection of *M. tuberculosis* and co-infecting pathogens directly from clinical specimens. However, compared with mNGS, tNGS enables targeted enrichment of known resistance-associated regions and clinically relevant pathogens, allowing for higher sequencing depth, improved sensitivity, and reduced host background, while retaining the ability to detect a broad range of resistance mutations and co-infecting organisms (4–7). The World Health Organization (WHO) supports the use of tNGS for DR-TB management due to its accuracy and efficiency (2). However, comprehensive assessments of tNGS performance across various pulmonary and extrapulmonary samples remain limited, indicating a need for further study (8).

This study aims to assess the diagnostic performance of tNGS in detecting TB, DR-TB, and co-infecting respiratory pathogens across a range of sample types, providing a thorough evaluation of its capabilities.

## MATERIALS AND METHODS

### Study population

We enrolled patients displaying symptoms indicative of tuberculosis at Kunming Third People's Hospital between November 2023 and February 2024. A total of 159 patients, from whom 126 pulmonary samples (e.g., sputum, bronchoalveolar lavage fluid, lung tissue) and 33 extrapulmonary samples (e.g., pus, cerebrospinal fluid [CSF], pleural effusion, and ascites) were collected (Table S1).

For each patient, the clinical specimen collected as part of routine TB diagnostics (respiratory or extrapulmonary) was included in this study, and all enrolled specimens underwent tNGS for pathogen detection and resistance monitoring. In parallel, we retrospectively retrieved TB testing results from the medical record system for the same specimens, including microbiological culture using the BACTEC MGIT 960 system (Becton, Dickinson and Company, USA), AFB smear microscopy using Ziehl-Neelsen staining, Xpert MTB/RIF assay using the G4 cartridge (Cepheid, USA), and TB-DNA testing by a real-time PCR kit for *M. tuberculosis* complex (Zhishan Biotech, China). In addition, we collected results of two interferon-γ (IFN-γ) release assays—T-SPOT.TB (Oxford Immunotec, UK) and QuantiFERON-TB Gold (QFT, Guangzhou Fenghua Biotech, China)—when available. Thus, all microbiological diagnostic methods were effectively applied to the same single specimen per patient, minimizing inter-sample heterogeneity.

Patients were categorized into three groups—confirmed TB, clinically diagnosed TB, and suspected TB—based on a composite reference standard that incorporated microbiological, clinical, radiological, and treatment response criteria, in accordance with the WHO guidelines and the Chinese National TB Diagnosis Standard (WS 288-2017). Confirmed TB referred to patients with positive results from any conventional microbiological test, including culture, GeneXpert MTB/RIF, AFB staining, or TB-DNA PCR, or with histopathological findings consistent with tuberculosis. Clinically diagnosed TB included patients who tested negative on microbiological assays but exhibited TB-compatible symptoms (e.g., persistent cough, fever, night sweats, weight loss), characteristic imaging findings, and showed a favorable response to anti-TB therapy, as evaluated by a panel of experienced TB clinicians. Suspected TB encompassed patients with incomplete diagnostic evidence who were evaluated for TB based on suggestive clinical or imaging findings but lacked both microbiological confirmation and sufficient clinical or radiological support for a definitive diagnosis. This classification ensured consistency in evaluating the diagnostic performance of tNGS across diverse clinical contexts, including both pulmonary and extrapulmonary TB.

## tNGS assay

In our investigation, we used a multiplex-PCR-based tNGS assay to analyze biological specimens from enrolled suspected TB patients (Supplemental methods). We employed the GenK magnetic bead-based DNA/RNA extraction kit for nucleic acid extraction and purification, followed by the GenK tuberculosis infection pathogen nucleic acid detection kit for amplification. Sequencing was performed on the MGISEQ-200 platform, with subsequent quality control and data processing steps to remove host DNA and identify pathogen-specific sequences. The tNGS output report included the presence or absence of *Mycobacterium tuberculosis* complex (MTBC) in the sample, drug resistance predictions for TB pathogens, and the detection results for common respiratory pathogens, including nontuberculous mycobacteria (NTM) (see Table S6 for detection list).

## Smear microscopy and culture

Researchers prepared slides directly from the specimens and examined them for AFB using Ziehl-Neelsen staining under a microscope. Processed samples were inoculated into the BD BACTEC MGIT 960 liquid culture system (Becton, Dickinson and Company, Franklin Lakes, NJ, USA), with parallel inoculation onto Löwenstein-Jensen (LJ) solid medium (Hopebio, Qingdao, China) as a complementary method. The LJ medium served as a backup when MGIT cultures were contaminated or yielded uninterpretable results. Cultures were incubated for up to 42 days, with observations made on the 3rd and 7th days post-inoculation and weekly thereafter until the end of the 6th week. The presence of *M. tuberculosis* was confirmed using the MPT64 (Abbott Diagnostics, Yongin, South Korea) antigen detection method, an immunochromatographic assay that detects the MPT64 protein secreted by members of the MTBC.

## GeneXpert MTB/RIF

GeneXpert MTB/RIF assays were conducted using the G4 cartridge as recommended by the manufacturer (Cepheid, Sunnyvale, California, USA). The test results provided information on the presence or absence of *M. tuberculosis* and indicated rifampicin resistance or susceptibility status. This assay detects mutations in the 81 bp rifampicin resistance-determining region (RRDR) of the rpoB gene, which is responsible for most rifampicin resistance-associated mutations. However, certain rare mutations outside the RRDR may not be captured, as noted in the WHO mutation catalog. This is a commercially available, automated molecular assay that has been widely adopted for rapid TB diagnosis from respiratory specimens.

## TB-DNA

The TB-DNA assay was performed using a real-time fluorescent PCR kit for the *M. tuberculosis* complex from Xiamen Zhishan Biotech Co., Ltd. Patient samples were prepared according to the manufacturer's protocol, and DNA was extracted and amplified using real-time PCR. Specific probes targeting MTBC DNA allowed for detection of the amplified product through fluorescence measurement.

## Interferon-γ release assays

Two IFN-γ release assays, QuantiFERON-TB Gold (QFT) and T-SPOT.TB, were used to detect *M. tuberculosis* infection. The QFT assay (Guangzhou Fenghua Biotech Co., Ltd.) involved incubating whole blood with ESAT-6, CFP-10, and TB7.7 antigens, followed by IFN-γ quantification using ELISA. Samples were divided into Mitogen (positive control), Nil (negative control), and TB antigen tubes, with IFN-γ measured within 24 hours.

The T-SPOT.TB assay (Oxford Immunotec Ltd.) utilized isolated peripheral blood mononuclear cells incubated with specific antigens (ESAT-6, CFP-10, and Rv3615c).

## Statistical analysis

Baseline demographic and clinical characteristics of the study participants were systematically evaluated. Continuous variables were presented as the median and interquartile range. Sensitivity, specificity, and overall accuracy were calculated for each diagnostic method. McNemar's chi-squared test was used to compare the sensitivity of tNGS with other methods, assessing the statistical significance of differences in performance.

## RESULTS

### Baseline clinical characteristics

From Kunming Third People's Hospital, we retrospectively enrolled 159 adult patients exhibiting suspected tuberculosis symptoms and radiographic characteristics (Fig. S1). According to the composite reference standard, 91 patients were diagnosed with PTB, 33 with EPTB, and 35 were classified as nontuberculous (NTB) (9). The NTB category included patients under follow-up for previous TB and those diagnosed with nontuberculous mycobacterial disease or other non-TB conditions (Fig. S2).

The clinical characteristics of these patients are summarized in Table 1. The overall age of the patients ranged from 9 to 83 years, with a median age of 51 years. Gender distribution showed that 38.4% ($n = 61$) of the overall cohort were female. The proportion of females was significantly higher ($P < 0.01$) in the EPTB group (63.6%, $n = 21$) compared to the PTB group (27.5%, $n = 25$) and the NTB group (42.9%, $n = 15$). The analysis of blood routine parameters (white blood cell count, WBC%; neutrophil percentage, NEU%; lymphocyte percentage, LYM%) and inflammatory indices (C-reactive protein; procalcitonin) showed no significant ($P > 0.05$) differences among the PTB, EPTB, and NTB groups (Table 1).

### tNGS demonstrates high consistency with CMTs and nucleic acid amplification tests (NAATs) in detecting TB pathogens

In our study, we compared the consistency of several TB pathogen diagnostic methods with the gold standard culture method (Fig. 1A; Table S2). Compared to microbial culture, the tNGS method demonstrated a positive percent agreement (PPA) of 91.7% (22/24), a negative percent agreement (NPA) of 73.9% (68/92), and an overall percent agreement (OPA) of 77.6% (90/116). In the confirmed TB group, tNGS failed to detect two cases that were positive by culture. However, these two cases were also negative by all other molecular methods, suggesting the possibility of low bacterial load or sampling limitations rather than a method-specific failure. Additionally, all the 15 samples that

**TABLE 1** Baseline characteristics of the study population[a,b,c]

|  | Overall | PTB | EPTB | NTB | *P*-value |
|---|---|---|---|---|---|
| **Basic Information** | | | | | |
| *n* | 159 | 91 | 33 | 35 | |
| Age (years, [range]) | 51 [9, 83] | 51 [11, 74] | 50 [9, 83] | 51 [24, 80] | |
| **Gender (%)** | | | | | |
| Female | 61 (38.4) | 25 (27.5) | 21 (63.6) | 15 (42.9) | <0.01 |
| Male | 98 (61.6) | 66 (72.5) | 12 (36.4) | 20 (57.1) | <0.01 |
| **Laboratory index** | | | | | |
| WBC ($10^9$/L) | 6.00 [4.47, 7.57] | 6.01 [4.43, 7.78] | 6.11 [4.71, 7.27] | 5.96 [4.52, 7.10] | 0.83 |
| NEU% | 64.30 [52.90, 74.60] | 64.35 [52.92, 74.52] | 66.00 [52.90, 76.10] | 56.65 [52.80, 74.12] | 0.80 |
| LYM% | 21.20 [11.60, 31.60] | 19.20 [11.98, 30.30] | 21.20 [13.10, 28.50] | 25.55 [9.02, 33.77] | 0.71 |
| ESR (mm/h) | 18.00 [6.00, 42.00] | 20.00 [6.00, 50.00] | 21.00 [12.75, 34.75] | 12.00 [3.00, 25.00] | 0.11 |
| **Inflammatory index** | | | | | |
| CRP (mg/L) | 11.61 [3.35, 45.67] | 15.10 [4.50, 53.30] | 9.14 [3.85, 20.44] | 7.80 [1.66, 40.35] | 0.34 |
| **PCT (ng/mL)** | | | | | |
| <0.05 | 101 | 56 | 23 | 22 | |
| >0.05 | 0.09 [0.05, 0.23] | 0.11 [0.05, 0.32] | 0.08 [0.05, 0.46] | 0.08 [0.05, 0.15] | 0.65 |

[a]Unless otherwise specified, the data in the table are presented as median [interquartile range].
[b]Group comparisons were performed using the Kruskal-Wallis test.
[c]PTB, pulmonary tuberculosis; EPTB, extrapulmonary tuberculosis; NTB, nontuberculous disease; WBC, white blood cell count; NEU%, neutrophil percentage; LYM%, lymphocyte percentage; ESR, erythrocyte sedimentation rate; CRP, C-reactive protein; PCT, procalcitonin.

tNGS detected but culture did not in the confirmed TB group were positive in either Xpert or TB-DNA tests.

Furthermore, we evaluated the consistency of the tNGS method with two NAATs, Xpert and TB-DNA, to assess the agreement between these methods (Table S3). For the overall group, when compared with Xpert, tNGS showed a PPA of 90.0%, an NPA of 85.7%, and an OPA of 86.8%. Similarly, when compared with TB-DNA, tNGS exhibited a PPA of 87.2%, an NPA of 84.2%, and an OPA of 85.2%.

## tNGS demonstrates superior clinical diagnostic performance over CMTs and NAATs in various sample types

We evaluated the clinical diagnostic performance of tNGS and compared it with several other tuberculosis pathogen detection methods across various diagnostic categories (confirmed TB, clinically diagnosed TB, and suspected TB) and sample types (BALF and lung tissue, sputum, extrapulmonary samples) (Fig. 1B; Table S4).

In a cohort of 85 BALF and 1 lung tissue sample, tNGS exhibited a positivity rate of 83.3% within the confirmed TB category, identifying 25 positive cases out of 30. This positivity rate of tNGS was notably higher than that achieved through culture (46.2%, 12/26), Xpert MTB/RIF (75%, 12/16), and AFB staining (27.6%, 8/29). TB-DNA, however, detected 26 positive cases from 29 samples, reaching the highest positivity rate of 89.7%. Additionally, tNGS detected seven more TB-positive cases than other methods in the clinically diagnosed and suspected TB groups.

In 40 sputum samples, tNGS recognized 15 out of 17 confirmed TB cases, achieving a positivity rate of 88.2%. The positivity rate of tNGS was superior to the positivity rates observed with sputum culture (68.8%, 11/16), TB-DNA (75%, 12/16), and markedly higher than AFB staining (35.3%, 6/17). All six sputum samples tested with Xpert showed positive results, yielding a positivity rate of 100%. In addition, tNGS detected four TB-positive cases in the clinically diagnosed and suspected TB groups that were negative by all conventional methods (culture, AFB staining, TB-DNA, and Xpert). These cases were ultimately classified as TB based on composite clinical evaluation, including consistent radiographic findings, discharge diagnosis, and favorable response to anti-TB therapy at one-month follow-up.

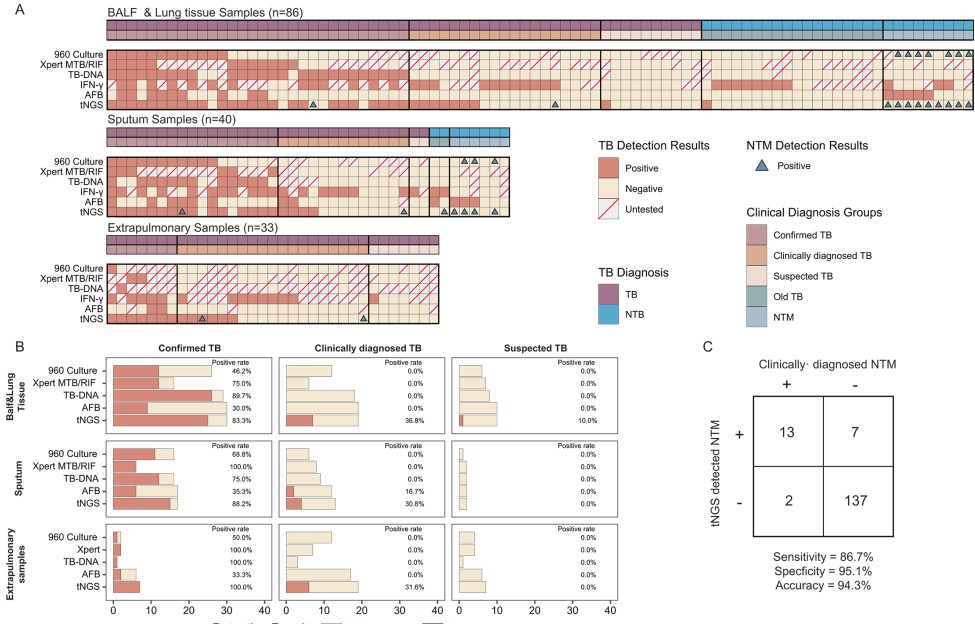

**FIG 1** Detection of TB pathogens in various samples using multiple diagnostic methods. Panel A shows the detection results of five different pathogen detection methods across all sample types, divided into three categories of sample types, and also indicates the detection of NTM by tNGS and culture. Panel B shows the positive rates of the five detection methods for different diagnostic groups and sample types. Panel C shows the diagnostic performance of tNGS for NTM. Abbreviations: BALF, bronchoalveolar lavage fluid; TB, tuberculosis; NTM, nontuberculous mycobacteria; AFB, acid-fast bacilli; tNGS, targeted next-generation sequencing; IFN, interferon-γ release assay.

In the 33 extrapulmonary samples, tNGS achieved a 100% positivity rate in the seven confirmed TB cases, identifying positives that were confirmed by other methods: one culture positive, two Xpert MTB/RIF positives, one TB-DNA positive, and two positive AFB stains. Among the clinically diagnosed EPTB specimens, tNGS detected MTBC in 6 of 19 cases that were negative or untested by conventional microbiological methods (Fig. 1A).

## tNGS detection performance for PTB and EPTB

In summary, in real clinical scenarios, for all samples, tNGS demonstrated a sensitivity of 52.4%, higher than other methods such as Xpert, which had a sensitivity of 34.5%, and TB-DNA, which had a sensitivity of 44.8%. Specifically, among the clinically diagnosed TB and suspected TB groups, tNGS detected nine additional cases in the PTB samples and six additional cases in the EPTB samples.

In the PTB group, using the composite reference standard, tNGS achieved a sensitivity of 57.1%, a specificity of 97.1%, and an accuracy of 68.3%. In comparison, Xpert showed a sensitivity of 40.0%, a specificity of 100%, and an accuracy of 57.1%, while TB-DNA had a sensitivity of 46.3%, a specificity of 100%, and an accuracy of 60.0%. In the EPTB group, tNGS identified 13 positive cases out of 33 samples, achieving a sensitivity of 39.4%. The performance of tNGS was superior to Xpert, which detected two positive cases out of 13 samples (15.4%), and TB-DNA, which found one positive case out of 5 samples (20.0%) (Table S4).

## Detection of drug resistance by tNGS

To assess the concordance between tNGS and Xpert for rifampicin resistance, we focused on 30 TB-positive specimens that were positive by either Xpert or tNGS and had valid results from both assays (Fig. 2A). Within this subset, Xpert reported rifampicin resistance in six cases, while tNGS detected rpoB resistance mutations in five of these six cases, yielding a positive percent agreement of 83.3% (5/6). In the same 30 specimens,

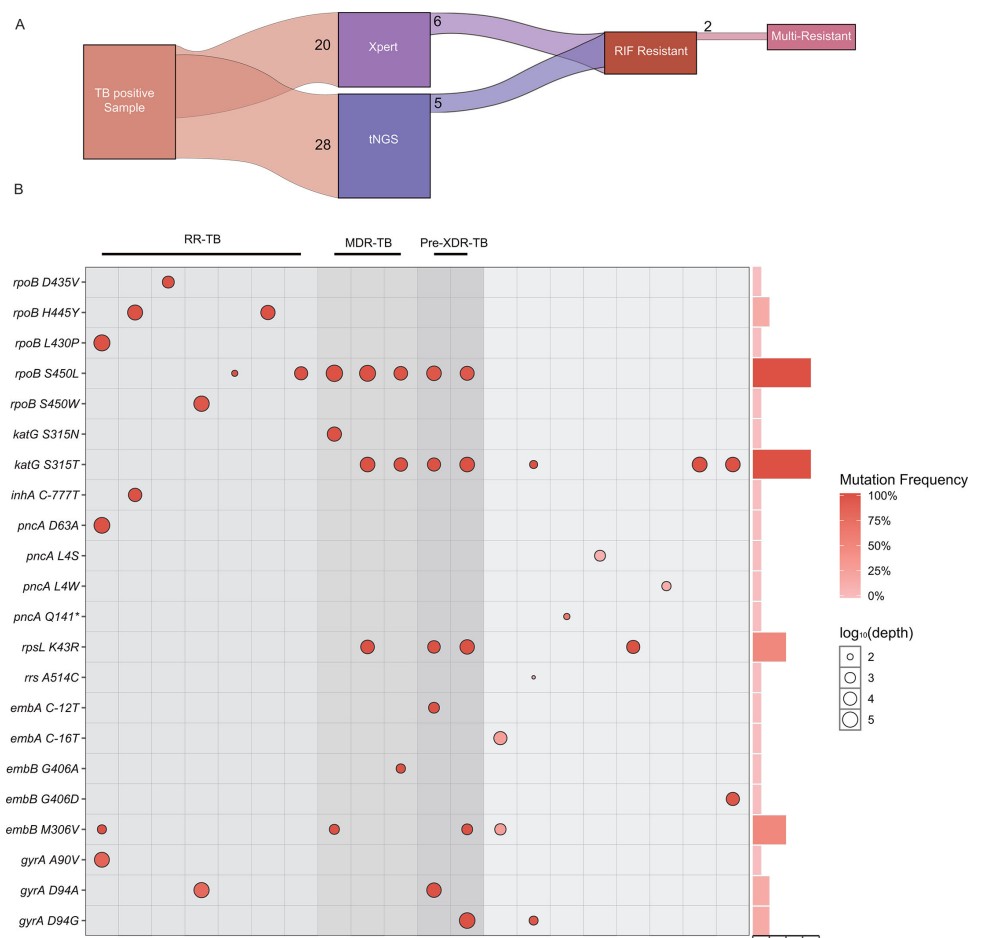

**FIG 2** Clinical diagnostic performance of tNGS for detecting drug resistance mutations. (A) This Sankey plot illustrates the consistency between tNGS and Xpert MTB/RIF in diagnosing rifampicin resistance. It also highlights the capability of tNGS to diagnose multi-drug resistance simultaneously. (B) The rect plot displays the resistance mutations detected in drug-resistant TB samples using tNGS in this study. The size of the dots represents the sequencing depth for each mutation site detected by tNGS, while the shades of red indicate the mutation frequency at each site. The bar chart on the right provides annotation for the positive detection frequency of each mutation site. Abbreviations: RIF, rifampicin; tNGS, targeted next-generation sequencing; TB, tuberculosis; RR-TB, rifampicin-resistant tuberculosis; MDR-TB, multidrug-resistant tuberculosis; pre-XDR-TB, pre-extensively drug-resistant tuberculosis.

tNGS also identified resistance mutations to drugs other than rifampicin (e.g., isoniazid, fluoroquinolones) in seven patients, which are not captured by Xpert.

When all tNGS-positive TB specimens in the cohort were considered, 20 patients harbored at least 1 of 22 distinct resistance-conferring mutation types (Fig. 2B). The most frequently observed mutations were rpoB S450L, associated with rifampicin resistance, in 12 instances, and katG S315N, associated with isoniazid resistance, in 9 instances. According to WHO case definitions, these resistance profiles corresponded to seven patients with rifampicin-resistant TB (RR-TB), including three with multidrug-resistant TB (MDR-TB) and two with pre-extensively drug-resistant TB (pre-XDR-TB).

## Co-detection of TB and other pathogens

In addition to detecting TB and MDR-TB, tNGS can also identify a range of common pathogens. Firstly, we evaluated the diagnostic performance of tNGS for NTM. According to the confusion matrix (a 2 × 2 table summarizing true positives, false positives, false negatives, and true negatives, Fig. 1C), tNGS demonstrated a sensitivity of 86.7%, a

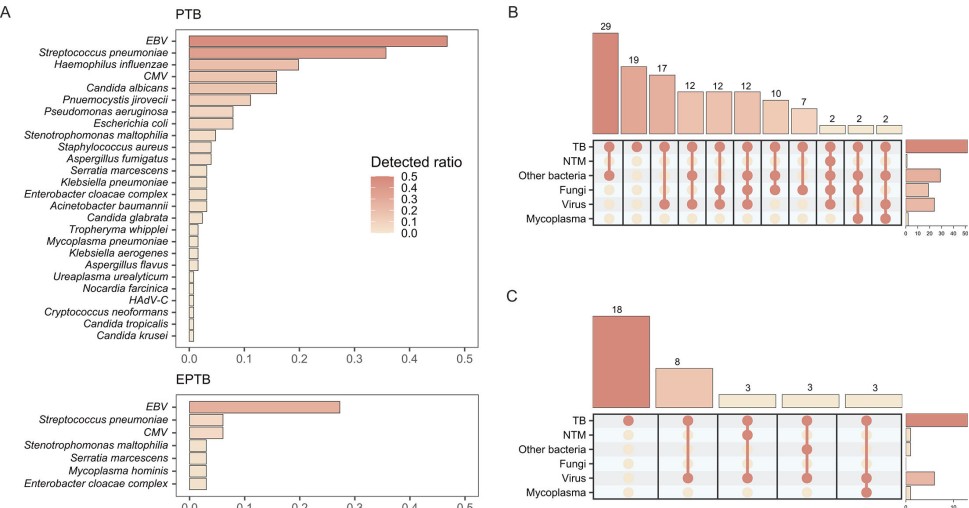

**FIG 3** Co-detection of TB and other pathogens. (A) Frequency of other pathogens detected in clinically diagnosed PTB and EPTB samples. (B) Subset plot of co-detection of TB and other pathogens in PTB samples. (C) Subset plot of co-detection of TB and other pathogens in EPTB samples. All pathogens shown were identified using the tNGS assay. Abbreviations: TB, tuberculosis; NTM, nontuberculous mycobacteria.

specificity of 95.1%, and an accuracy of 94.3%. The detection rates of NTM subtypes are summarized in Table S5. *Mycobacterium avium complex* was identified in 8.2% of cases, followed by *M. chelonae* (3.1%), *M. abscessus* (1.3%), and *M. kansasii* (0.6%).

For samples already confirmed to have TB, we examined the co-detection of *M. tuberculosis* complex with other pathogens. In samples diagnosed as PTB and EPTB, EBV, *Streptococcus pneumoniae,* and CMV were detected at relatively high rates (Fig. 3A). *Haemophilus influenzae,* which was frequently detected in PTB samples, was not detected in EPTB samples. In PTB samples (Fig. 3B), the co-detection rate of MTBC with other bacterial pathogens was 23.1%, higher than the rate of TB detection alone (15.4%). The co-detection rate of MTBC with viruses was 13.5%. In contrast, in EPTB samples, MTBC was detected alone in 7 of the 13 samples (54%), and MTBC co-detection with viruses was observed in 23% of the samples. Additionally, co-detection of MTBC and NTM was observed in one PTB sample and one EPTB sample.

## DISCUSSION

Accurate and timely detection of TB and MDR-TB is crucial for effective treatment (2). This study retrospectively evaluated the clinical diagnostic performance of tNGS in various sample types for TB. Compared to traditional methods, tNGS demonstrated excellent diagnostic performance across different sample types from both PTB and EPTB cases. Additionally, tNGS can predict TB drug resistance through genotypic drug susceptibility testing and detect a wide range of common co-infecting pathogens alongside TB.

Our findings demonstrate that tNGS provides superior diagnostic performance across different types of samples. For PTB samples, mainly including BALF and sputum, tNGS achieved a higher sensitivity compared to Xpert MTB/RIF and TB-DNA, effectively identifying additional positive cases. The diagnosis of EPTB patients is often more challenging due to the affected sites and the difficulty in obtaining samples (10–12). In the EPTB specimens, including pus, CSF, etc., tNGS also demonstrated high sensitivity, especially in the confirmed TB group. This highlights the diagnostic value of tNGS for TB across various sample types. Unlike the Xpert MTB/RIF assay, which is validated primarily for respiratory specimens and has limited utility for extrapulmonary TB, tNGS imposes no such restriction on specimen type. This versatility enables comprehensive detection of Mycobacterium tuberculosis and resistance mutations from diverse clinical samples, including extrapulmonary sites, thereby broadening its potential diagnostic application.

tNGS showed superior performance in detecting TB pathogens compared to traditional and molecular diagnostic methods. In the confirmed TB group, tNGS showed high sensitivity. In clinically diagnosed TB and suspected TB groups, where the microbial load is often low in the samples, tNGS identified additional TB cases that were missed by culture and other molecular tests. The high consistency between tNGS and culture, combined with its ability to detect additional TB cases in clinically diagnosed and suspected TB groups, highlights its value in improving diagnostic accuracy and patient management (13).

In terms of drug resistance detection, tNGS demonstrated comparable detection of rifampicin resistance to Xpert. Moreover, tNGS can identify a wider array of resistance mutations, including those beyond the detection scope of Xpert MTB/RIF, offering a more comprehensive resistance profile (5, 6, 14). The detection of 22 resistance mutations in 20 samples, with *rpoB* S450L and *katG* S315N being the most frequent, is consistent with WHO's statistics on the high prevalence of these two mutations (15, 16). In this cohort, tNGS detected resistance mutations beyond the rifampicin resistance reported by Xpert, covering isoniazid, ethambutol, streptomycin, fluoroquinolones, and second-line injectable agents. These findings, aligned with WHO recommendations, highlight tNGS's capacity to identify cases of RR-TB, MDR-TB, and pre-XDR-TB, making it a crucial tool for guiding appropriate treatment strategies.

The detection of other pathogens has substantial clinical significance, as it may guide appropriate empirical treatment, reveal co-infection patterns that influence clinical outcomes, and help distinguish TB from other infectious causes of similar pulmonary symptoms (15, 17). In our cohort, tNGS showed high sensitivity and specificity for detecting NTM. It also detected pathogens such as EBV, Streptococcus pneumoniae, and CMV alongside *M. tuberculosis*, indicating that tNGS can capture potential co-infections within a single assay. In our cohort, several illustrative cases underscored this potential: in one patient with radiographic evidence of lung cavitation, tNGS revealed co-infection with *Aspergillus fumigatus*, prompting initiation of antifungal therapy and subsequent clinical improvement; in another patient, co-detection of *Nocardia cyriacigeorgica* led to the administration of sulfamethoxazole, with radiologic improvement on follow-up imaging; and in three additional cases in which clinicians had already suspected fungal co-infection, tNGS confirmation of fungal pathogens such as *Candida albicans* supported the continuation of antifungal therapy. Although these observations derive from a small, non-systematically ascertained subset and our study was not designed to formally evaluate outcome differences, they illustrate how the ability of tNGS to detect multiple pathogens in one step can streamline diagnostic processes, support the recognition of clinically meaningful co-infection patterns involving *M. tuberculosis* and other microbial agents, help clarify atypical presentations, and inform more individualized, holistic patient care.

Despite its advantages, tNGS has several limitations that warrant consideration. First, the relatively high cost of tNGS and its requirement for specialized laboratory infrastructure remain important barriers to widespread implementation, particularly in resource-limited settings. Although cost-sharing strategies, such as centralized laboratories, may help mitigate these issues, broader accessibility will require further simplification and cost reduction of workflows. Second, tNGS may have reduced sensitivity in specimens with low bacterial burden, where traditional culture retains an advantage. Additionally, tNGS generally has a longer turnaround time compared to rapid molecular assays like Xpert, which can provide results within a few hours. Pre-analytical factors, such as sample type, processing quality, and DNA yield, can also affect detection performance and were not controlled in this retrospective study. Third, while our study aimed to evaluate diagnostic performance in a real-world clinical setting, its retrospective and single-center design inherently limits the generalizability of the findings. The use of non-uniform testing across patients (e.g., not all underwent culture, Xpert, and TB-DNA) reduces the comparability of results across methods and may introduce bias. The limited sample size, especially in subgroups such as extrapulmonary TB, further constrains the statistical

power and robustness of sensitivity and specificity estimates. Finally, although resistance mutations were interpreted based on established databases, we did not perform an in-depth description of the bioinformatic algorithms used for resistance calling. Future studies should include comprehensive evaluation of bioinformatics pipelines, validation in multicenter prospective cohorts, and external benchmarking with gold-standard phenotypic testing.

## ACKNOWLEDGMENTS

This research was funded by the State Key Laboratory of Pathogen and Biosecurity Open Research Project (SKLPBS2445), the Yunnan Province Major Science and Technology Special Plan (202402AA310011), the Clinical Medical Center Scientific Research Project of Yunnan Province (2024YNLCYXZX0205), and the Kunming Science and Technology Bureau (2023-1-NS-007).

Lifeng Li, Gang Zhou, and Fengshuo Sun are employed by Genskey Inc., Beijing, China. The authors declare that they have no competing interests.

## AUTHOR AFFILIATIONS

[1]Department of Public Laboratory, The Third People's Hospital of Kunming City/Infectious Disease Clinical Medical Center of Yunnan Province, Kunming, Yunnan, People's Republic of China

[2]International Research Fellow, Prince of Songkla University, Hat Yai, Songkhla, Thailand

[3]Genskey Medical Technology Co., Ltd, Beijing, People's Republic of China

[4]College of Life Sciences, Peking University, Beijing, People's Republic of China

[5]Department of Medical Laboratory, The Third People's Hospital of Kunming City/ Infectious Disease Clinical Medical Center of Yunnan Province, Kunming, Yunnan, People's Republic of China

## AUTHOR ORCIDs

Jingyi Dai http://orcid.org/0000-0002-9210-8902
Qiujing Li http://orcid.org/0009-0005-6817-9240
Fengshuo Sun https://orcid.org/0009-0005-7666-7113
Lin Wang http://orcid.org/0000-0001-8008-6104
Guiming Liu http://orcid.org/0009-0008-7433-7075

## FUNDING

| Funder | Grant(s) | Author(s) |
| --- | --- | --- |
| State Key Laboratory of Pathogen and Biosecurity open research project | SKLPBS2445 | Jingyi Dai |
| Yunnan Province Major Science and Technology Special Plan | 202402AA310011 | Jingyi Dai |
| Clinical Medical Center Scientific Research Project of Yunnan Province | 2024YNLCYXZX0205 | Jingyi Dai |
| Kunming Science and Technology Bureau | 2023-1-NS-007 | Jingyi Dai |

## AUTHOR CONTRIBUTIONS

Jingyi Dai, Investigation, Writing – original draft | Qiujing Li, Data curation, Writing – original draft, Writing – review and editing | Juan Wan, Formal analysis, Investigation | Fengshuo Sun, Formal analysis, Visualization | Gang Zhou, Methodology, Writing – original draft | Meiqiu Yang, Resources | Chuanshu Dong, Validation | Yao Fang, Formal analysis | Lifeng Li, Conceptualization, Project administration, Writing – review and editing | Lin Wang, Supervision, Writing – review and editing | Guiming Liu, Conceptualization, Resources, Supervision, Writing – original draft, Writing – review and editing

## DATA AVAILABILITY

The raw sequencing data generated in this study have been deposited in the Genome Sequence Archive (GSA) under BioProject accession number PRJCA047391.

## ETHICS APPROVAL

This study was conducted in accordance with the Declaration of Helsinki (2013 revision) and approved by the Ethics Committee of Kunming Third People's Hospital (Ethics Approval Number: KSLL2023071151). Informed consent was obtained in writing from all participants prior to their involvement in the study.

## ADDITIONAL FILES

The following material is available online.

### Supplemental Material

**Supplemental material (Spectrum01698-25-s0001.docx).** Supplemental methods; Fig. S1 and S2; Tables S1 to S6.

### Open Peer Review

**PEER REVIEW HISTORY (review-history.pdf).** An accounting of the reviewer comments and feedback.

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
