## [Reviewer comments · Microbiology Spectrum]

Microbiology Spectrum

Targeted Next-Generation Sequencing for Comprehensive Diagnosis and Drug Resistance Detection in Pulmonary and Extrapulmonary Tuberculosis: A Single-Center Retrospective Study

Jingyi Dai, Qiuqing Li, Juan Wan, Fengshuo Sun, Gang Zhou, Meiqu Yang, Chuanshu Dong, Yao Fang, Lifeng Li, Lin Wang, and Guiming Liu

Corresponding Author(s): Guiming Liu, The Third People's Hospital of Kunming

Review Timeline:

Submission Date:	June 3, 2025
Editorial Decision:	August 6, 2025
Revision Received:	October 6, 2025
Editorial Decision:	October 27, 2025
Revision Received:	December 17, 2025
Accepted:	January 13, 2026

Editor: Kenneth Gavina

Reviewer(s): The reviewers have opted to remain anonymous.

Transaction Report:

DOI: <https://doi.org/10.1128/spectrum.01698-25>

Re: Spectrum01698-25 (**Targeted Next-Generation Sequencing for Comprehensive Diagnosis and Drug Resistance Detection in Pulmonary and Extrapulmonary Tuberculosis: A Single-Center Retrospective Study**)

Dear Dr. Guiming Liu:

Thank you for the privilege of reviewing your work. Your manuscript has been selected for re-submission with modifications. Below you will find the comments and instructions from the reviewers, along with instructions from the Spectrum editorial office. Please ensure to address all points and comments before resubmission.

Revision Guidelines

Sincerely,
Kenneth Gavina
Editor
Microbiology Spectrum

Reviewer #1 (Comments for the Author):

This paper describes a targeted next generation sequencing assay to detect *Mycobacterium tuberculosis* and key drug resistance mutations. The assay was compared to other assays currently used to diagnose TB infections. In addition, the assay has the ability to detect other important respiratory pathogens.

Major Comments:

- 1) Line 51, Lines 90-97: The ability to identify co-infections using targeted NGS depends on the primers used. The authors do not discuss the primers used for their assay, nor do they discuss which organisms can or cannot be detected. This is critically important to at least include what the target is (even if the specific sequence is not listed), so that the readers know which organisms can/cannot be detected.
- 2) Figure 2: The inclusion of the resistance markers is helpful, but I think the authors should evaluate the performance of this component of the assay against strains with known resistance, across the entire spectrum of resistance. Additionally, the authors should clarify if these resistance markers agreed with susceptibility testing/clinical response to therapy.

Minor Comments:

- 1) Lines 28-29: Repetitive comments. Please condense.
- 2) Line 30: Should be prospectively instead of retrospectively.
- 3) Line 33: PTB and EPTB have not been defined yet, so please define before using the acronyms.
- 4) Lines 48-51: Repetitive comments regarding the diagnostic challenge for detecting TB. Please revise to make this more concise.
- 5) Lines 53-55: Next generation sequencing is still very expensive and labor intensive, so I'm not sure how this assay would necessarily help in underserved settings.
- 6) Line 64: *Mycobacterium tuberculosis* should be italicized.
- 7) Line 66: Can you expand on "specific genome regions" with examples? See Major Comment 1.
- 8) Lines 68-69: What is tNGS being compared to for this sentence? True mNGS would be superior in terms of what can be detected.
- 9) Line 84: Is "Acid Fast Bacteria (AFB)" referring to a stain (such as Ziehl-Neelsen)? If so, please specify.
- 10) Line 84: Define Xpert MTB/RIF (assay and company).
- 11) Line 84: Define TB-DNA (assay and company).
- 12) Line 85: Define T-SPOT (assay and company).
- 13) Line 85: Define QFT (assay and company).
- 14) Lines 96-97: What are some examples of common respiratory pathogens? See Major Comment 1.
- 15) Lines 100-101: Were samples cultured on any other growth medium (such as Löwenstein-Jensen agar) or only the liquid MGIT broth?
- 16) Lines 104-107: Authors should add a comment indicating that this is a commercially available assay commonly used to diagnose TB from respiratory specimens.
- 17) Line 182: Prospectively instead of retrospectively. See Minor Comment 2.
- 18) Lines 129-130: Please confirm that all patients only submitted one sample. For TB rule out, CDC recommends that "Patients should have at least three consecutive sputum specimens examined, each collected in 8 to 24-hour intervals (at least one collected early in the morning)." (<https://www.cdc.gov/tb/hcp/testing-diagnosis/clinical-and-laboratory-diagnosis.html>)
- 19) Lines 141-142: Please include the number of samples in addition to the percentage (n=#).
- 20) Lines 142-144: Indicate that these results can be found in Table 1.
- 21) Lines 149-150: I do not believe this sentence is saying what it is intended to say. I think this is saying that there were two cases that were detected in culture but not by tNGS, but the number of cases detected by tNGS is still higher than by culture. However, the way it is worded makes it seem like tNGS detected two fewer TB cases when compared to culture, which is not true. Please revise for clarity.
- 22) Lines 154-156: Are any of these differences statistically significant?
- 23) Lines 163-164: How was this cohort chosen?
- 24) Throughout: Please spell out numbers less than ten (one instead of 1).
- 25) Lines 163-164: I don't necessarily agree with only looking at sensitivity and specificity within the TB positive patients. This could be used in comparison with total sensitivity and specificity, but the intent of the assay is to detect TB in ANY patient.
- 26) Lines 169-170: Why are the denominators different (17 vs 16)?
- 27) Line 184: 40% sensitivity should not be described as "high sensitivity."
- 28) Lines 186-187: Extrapulmonary TB specimen sources would be considered "off-label" use of the assay. This should be included in the discussion.
- 29) Line 191: "Xpert and tNGS" instead of "Xpert or tNGS?"
- 30) Line 202: Define "confusion matrix."
- 31) Lines 203-205: This sentence needs to be revised.
- 32) Line 204: *Mycobacterium avium* should be italicized.
- 33) Line 208: "However" can be removed from the beginning of this sentence.
- 34) Line 210: "Following this," can be removed from the beginning of this sentence.
- 35) Line 224: "Etc." instead of "et al."?
- 36) Line 240: Please describe how detection of other pathogens has a "substantial clinical significance."
- 37) Line 247: How does tNGS enhance the understanding of TB's interaction with other infectious agents? Please clarify.
- 38) Table 1: For the p-value, what is being compared to what?
- 39) Figure 1C: "Clinically" instead of "Clinical."
- 40) Figure 3: Please list how the other pathogens were detected? tNGS? Culture? Multiple methods?
- 41) Figure 3: Would prefer actual numbers instead of a detected ratio.
- 42) Figure 3: Why is *Mycoplasma* separated out from other bacteria?

43) Supplemental methods: What 64 species were included in this panel? Can you include this as a table? See Major Comment 1.

44) Supplemental Table 1: It would be helpful to know the positivity rate for each sample type. Could you include this in the table?

Reviewer #2 (Comments for the Author):

Overview: Dai et al. addresses a relevant public health issue, TB, and use of targeted next-generation sequencing (tNGS). Comparative analysis with conventional methods and Xpert MTB/RIF is scientifically sound. As well as reporting of sensitivity percentages for different sample types adds value.

Minor:

Line 28 and 29 - Say effectiveness twice under the same context - can be combined.

Line 51 - Difficult to diagnose extrapulmonary specimens - is this accurate? It's the suspicion of extra pulmonary TB that is challenging not necessarily the sensitivity and specificity of testing in extrapulmonary sites.

Line 84-85 - TB-DNA, T-SPOT and QFT need to be defined prior to using short forms.

Line 103 - Define what the MPT64 antigen detection method is

Line 105-107 - May be worth stating or at least referencing that the Xpert uses the 81-bp RRDR region for determining resistance and even though it can catch most but not all Rifampicin resistant TB.

Line 134 - PTB and EPTB were not defined previously prior to using

Line 135 - What diagnostic guidelines?

Line 142 - 144 - Either in the table or within the paper you may want to define WBC%, NEU% etc.

Line 161 - Extrapulmonary does not need the capital E

Line 191 - seems to be an error "with at least one sample testing positive for TB"

Major:

Line 163 - 85 BALF and 1 tissue sample are these all from unique patients.

Line 163 - 168 - Inconsistent denominators for all of the testing. The variable sample set is a limitation of retrospective studies, however it is challenging to truly gather sensitivity or specificity in such a manner. It may make sense to normalize to one test group - all positives by any method vs tNGS

Line 164 - What is the confirmed TB category - how is this defined?

Line 167 - 168 - what is the correlate for saying tNGS detected 7 more TB cases? How can you be sure they are not false positives given all other tests were negative?

Line 163 -168 - Insufficient statistical support given the lack of p-values and PPV or NPV.

Line 169-173 - Similar issues with lack of consistency in denominators with a limited sample size. Unclear what defines "4 extra positives"

Line 174-177 - Similar issues with lack of consistency in denominators and how "extra positives" were confirmed.

Line 183 - 184 - Verify accuracy calculation of 68.3% - it is a bit odd given the high specificity and moderate sensitivity

Line 186 - 188 - Was the Xpert validated for the extrapulmonary sites?

Line 190-198 - Given that Xpert can only detect Rifampicin resistance, it would be helpful to know whether the additional detected resistance to Isoniazid and MDR and XDR is consistent with local epidemiology.

Line 199-213 - Were the co-detection of other pathogens in these cases clinically relevant?

Line 249 - Limitations of tNGS were listed here but not the limitations of the paper regarding a retrospective study, inconsistent testing across specimens, limited sample size etc.

Overall, the paper doesn't really discuss the limitations of tNGS in terms of the high cost for the assay which is a large limiting factor. Sensitivity issues with detecting low organism burden which is really why culture really remains the gold standard. Or it's limitation with turnaround time which is the advantage of assays like Xpert. It also doesn't discuss limitations of pre-analytical factors like the specimen quality etc. Additionally, that it is a single center retrospective study that may limit generalizability of the findings.

The paper also doesn't discuss the bioinformatic tools and evaluation needed for determining resistance.

Claims in this study in terms of tNGS sensitivity does not seem sufficient and are overstated without consistent testing.

Review for: **Targeted Next-Generation Sequencing for Comprehensive Diagnosis and Drug Resistance Detection in 2 Pulmonary and Extrapulmonary Tuberculosis: A Single-Center Retrospective Study**

Overview: Dai et al. addresses a relevant public health issue, TB, and use of targeted next-generation sequencing (tNGS). Comparative analysis with conventional methods and Xpert MTB/RIF is scientifically sound. As well as reporting of sensitivity percentages for different sample types adds value.

Minor:

Line 28 and 29 – Say effectiveness twice under the same context – can be combined.

Line 51 – Difficult to diagnose extrapulmonary specimens – is this accurate? It's the suspicion of extra pulmonary TB that is challenging not necessarily the sensitivity and specificity of testing in extrapulmonary sites.

Line 84-85 – TB-DNA, T-SPOT and QFT need to be defined prior to using short forms.

Line 103 – Define what the MPT64 antigen detection method is

Line 105-107 – May be worth stating or at least referencing that the Xpert uses the 81-bp RRDR region for determining resistance and even though it can catch most but not all Rifampicin resistant TB.

Line 134 – PTB and EPTB were not defined previously prior to using

Line 135 – What diagnostic guidelines?

Line 142 – 144 – Either in the table or within the paper you may want to define WBC%, NEU% etc.

Line 161 – Extrapulmonary does not need the capital E

Line 191 – seems to be an error “with at least one sample testing positive for TB”

Major:

Line 163 – 85 BALF and 1 tissue sample are these all from unique patients.

Line 163 – 168 – Inconsistent denominators for all of the testing. The variable sample set is a limitation of retrospective studies, however it is challenging to truly gather sensitivity or specificity in such a manner. It may make sense to normalize to one test group – all positives by any method vs tNGS

Line 164 – What is the confirmed TB category – how is this defined?

Line 167 – 168 – what is the correlate for saying tNGS detected 7 more TB cases? How can you be sure they are not false positives given all other tests were negative?

Line 163 -168 – Insufficient statistical support given the lack of p-values and PPV or NPV.

Line 169-173 – Similar issues with lack of consistency in denominators with a limited sample size. Unclear what defines “4 extra positives”

Line 174-177 – Similar issues with lack of consistency in denominators and how “extra positives” were confirmed.

Line 183 – 184 – Verify accuracy calculation of 68.3% - it is a bit odd given the high specificity and moderate sensitivity

Line 186 – 188 – Was the Xpert validated for the extrapulmonary sites?

Line 190-198 – Given that Xpert can only detect Rifampicin resistance, it would be helpful to know whether the additional detected resistance to Isoniazid and MDR and XDR is consistent with local epidemiology.

Line 199-213 – Were the co-detection of other pathogens in these cases clinically relevant?

Line 249 – Limitations of tNGS were listed here but not the limitations of the paper regarding a retrospective study, inconsistent testing across specimens, limited sample size etc.

Overall, the paper doesn't really discuss the limitations of tNGS in terms of the high cost for the assay which is a large limiting factor. Sensitivity issues with detecting low organism burden which is really why culture really remains the gold standard. Or it's limitation with turnaround time which is the advantage of assays like Xpert. It also doesn't discuss limitations of pre-analytical factors like the specimen quality etc. Additionally, that it is a single center retrospective study that may limit generalizability of the findings.

The paper also doesn't discuss the bioinformatic tools and evaluation needed for determining resistance.

Claims in this study in terms of tNGS sensitivity does not seem sufficient and are overstated without consistent testing.

Dear Editor,

We sincerely thank you and the reviewers for the thoughtful and constructive comments on our manuscript entitled “*Targeted Next-Generation Sequencing for Comprehensive Diagnosis and Drug Resistance Detection in Pulmonary and Extrapulmonary Tuberculosis: A Single-Center Retrospective Study*” (Manuscript ID: Spectrum01698-25).

We have carefully revised the manuscript in accordance with all suggestions. Each reviewer’s comment has been addressed point by point in this **Response to Reviewers** document, and corresponding changes have been highlighted in the revised manuscript. These revisions have substantially improved the clarity and scientific rigor of our study.

In addition, we have made all raw sequencing data publicly available in the National Genomics Data Center (NGDC) under BioProject accession number **PRJCA047391**, in full compliance with ASM’s data availability requirements.

We sincerely appreciate the reviewers’ valuable input and your consideration of our revised submission. We hope that the revised version meets the journal’s standards for publication.

With kind regards,

Guiming Liu

(on behalf of all authors)

Response to Reviewers

Reviewer #1 (Comments for the Author):

This paper describes a targeted next generation sequencing assay to detect *Mycobacterium tuberculosis* and key drug resistance mutations. The assay was compared to other assays currently used to diagnose TB infections. In addition, the assay has the ability to detect other important respiratory pathogens.

Major Comments:

1) Line 51, Lines 90-97: The ability to identify co-infections using targeted NGS depends on the primers used. The authors do not discuss the primers used for their assay, nor do they discuss which organisms can or cannot be detected. This is critically important to at least include what the target is (even if the specific sequence is not listed), so that the readers know which organisms can/cannot be detected.

Response: Thank you for this important comment. We have now added a detailed description of the targeted pathogen panel and primer coverage in the **Supplementary Table 6**. Specifically, our tNGS panel was designed to target 64 clinically relevant pathogens, including *M. tuberculosis* and selected bacterial, fungal, and viral respiratory pathogens.

2) Figure 2: The inclusion of the resistance markers is helpful, but I think the authors should evaluate the performance of this component of the assay against strains with known resistance, across the entire spectrum of resistance. Additionally, the authors should clarify if these resistance markers agreed with susceptibility testing/clinical response to therapy.

Response:

Thank you for this valuable comment. We carefully reviewed all available phenotypic drug susceptibility testing (pDST) data for the included samples. Among the cohort, only three samples had complete pDST results available, and all three showed full concordance between the phenotypic resistance profiles and the tNGS-predicted resistance mutations. Given the small number of samples with pDST data, a statistical evaluation of tNGS performance against the phenotypic gold standard was not feasible.

For the remaining cases with tNGS-detected resistance mutations, we examined their clinical treatment records. We found that tNGS results were consistent with clinical management—patients with tNGS-detected rifampicin or isoniazid resistance were

switched to second-line regimens, following national TB treatment guidelines. This suggests that tNGS results provided clinically meaningful guidance for therapeutic decision-making, even when phenotypic testing was unavailable.

Based on the current dataset, we have briefly described this observation in the Discussion section (see Lines 278–282).

Minor Comments:

1) Lines 28-29: Repetitive comments. Please condense.

Response: Thank you for pointing this out. We have revised the sentence for conciseness and clarity. **(Line 29-29)**

2) Line 30: Should be prospective instead of retrospective.

Response: Thank you for pointing this out. This is a retrospective research. The term has been corrected from “recruited” to “enrolled.” **(Line30)**

3) Line 33: PTB and EPTB have not been defined yet, so please define before using the acronyms.

Response: We appreciate the suggestion. Definitions for PTB and EPTB have been added at first mention. **(Lines 55-56)**

4) Lines 48-51: Repetitive comments regarding the diagnostic challenge for detecting TB. Please revise to make this more concise.

Response:

Thank you for this helpful suggestion. We have revised the paragraph to avoid redundancy and improve clarity. **(Lines 47-49)**

5) Lines 53-55: Next generation sequencing is still very expensive and labor intensive, so I'm not sure how this assay would necessarily help in underserved settings.

Response:

Thank you for raising this important point. We agree that cost is a key concern for the implementation of NGS-based diagnostics in resource-limited settings. To address this, we have added a paragraph in the Discussion section estimating the patient-level cost of tNGS in our setting, and we highlight that with increasing adoption, costs are expected to decline. Furthermore, we propose a hub-and-spoke model for deployment,

where centralized laboratories serve multiple institutions via sample referral, allowing shared use of equipment and personnel to reduce per-sample costs. (See revised Discussion, **Lines 289-293**)

6) Line 64: *Mycobacterium tuberculosis* should be italicized.

Response: Thank you. Italicization has been corrected. (**Line 62**)

7) Line 66: Can you expand on "specific genome regions" with examples? See Major Comment 1.

Response:

Thank you for this helpful comment. We have revised the text to provide examples of the target genomic regions included in the tNGS panel, such as *rpoB*, *katG*, *inhA* promoter, *gyrA*, *gyrB*, and *embB*. Additionally, we have added a supplementary table listing all resistance-associated regions covered by the assay. (See **Line 113 and new Supplementary Table 6**)

8) Lines 68-69: What is tNGS being compared to for this sentence? True mNGS would be superior in terms of what can be detected.

Response:

Thank you for pointing this out. To address this, we have added a background sentence about the use of mNGS for TB and pathogen detection and clarified that tNGS offers a targeted and more sensitive alternative with lower host background interference. The revised paragraph now provides a clearer comparison between mNGS and tNGS in the context of clinical application. (See **Lines 66-71**)

9) Line 84: Is "Acid Fast Bacteria (AFB)" referring to a stain (such as Ziehl-Neelsen)? If so, please specify.

Response:

Thank you for these suggestions. We have revised the main text to define all mentioned assays, including the full test names and their respective manufacturers. These include the GeneXpert MTB/RIF assay (Cepheid, USA), TB-DNA PCR kit (Zhishan Biotech, China), T-SPOT.TB (Oxford Immunotec, UK), and

QuantiFERON-TB Gold (Guangzhou Fenghua Biotech, China). (See revised methods
Lines 98-100)

14) Lines 96-97: What are some examples of common respiratory pathogens? See Major Comment 1.

Response:

Thank you for your comment. We have now included representative examples of common respiratory pathogens in the main text and have provided the full list of 64 targeted organisms covered by the tNGS panel in a new Supplementary Table (Supplementary Table 6). (See revised Lines 113 and Supplementary Table 6)

15) Lines 100-101: Were samples cultured on any other growth medium (such as Löwenstein-Jensen agar) or only the liquid MGIT broth?

Response:

Thank you for this insightful question. In addition to the BACTEC MGIT 960 liquid culture system, we also performed solid culture using Löwenstein-Jensen (LJ) medium as a complementary method. LJ culture served as a backup in cases where MGIT cultures were contaminated or yielded ambiguous results. We have clarified this in the revised Methods section. (See **Lines 117-120**)

16) Lines 104-107: Authors should add a comment indicating that this is a commercially available assay commonly used to diagnose TB from respiratory specimens.

Response:

Thank you for the suggestion. We have added a sentence to clarify that GeneXpert MTB/RIF is a commercially available molecular assay that has been widely implemented for the diagnosis of *M. tuberculosis* and rifampicin resistance in respiratory specimens. (See revised **Lines 131-132**)

17) Line 182: Prospectively instead of retrospectively. See Minor Comment 2.

Response:

Thank you for pointing this out. You are correct that this study is retrospective in design. We previously misused the term “recruited” in this context. We have now corrected the sentence to accurately state that patients were “retrospectively included” based on predefined inclusion criteria.

18) Lines 129-130: Please confirm that all patients only submitted one sample. For TB rule out, CDC recommends that "Patients should have at least three consecutive sputum specimens examined, each collected in 8 to 24-hour intervals (at least one collected early in the morning)."

<https://www.cdc.gov/tb/hcp/testing-diagnosis/clinical-and-laboratory-diagnosis.html>

Response:

Thank you for raising this important point. We fully acknowledge the CDC recommendation of collecting three consecutive sputum specimens to enhance the sensitivity of conventional tests such as smear microscopy. However, as this study is based on a retrospective real-world cohort from routine clinical practice (November 2023 to February 2024), all samples included were single residual specimens obtained from standard diagnostic workflows. To minimize variability and ensure scientific comparability between methods, we implemented several measures:

1. All diagnostic methods (tNGS, culture, smear, etc.) were performed on the same specimen for each patient to eliminate inter-sample heterogeneity.
2. All tests were conducted in the same laboratory using standardized protocols and reagent batches to reduce procedural bias.
3. Patient classification was based on comprehensive clinical evaluation by at least three experienced TB specialists, rather than single-test results.

While we recognize the limitations of single-sample testing, our study reflects real-world diagnostic conditions and provides valuable evidence on the relative advantages of tNGS in such settings.

19) Lines 141-142: Please include the number of samples in addition to the percentage (n=#).

Response:

Thank you for your suggestion. We have added the corresponding number of samples (n=) alongside each percentage to improve clarity and transparency. (See revised **Lines 166-169**)

20) Lines 142-144: Indicate that these results can be found in Table 1.

Response:

Thank you for the suggestion. We have now added a reference to Table 1 in the relevant sentence to guide the reader to the corresponding detailed data. (See **Line 172**)

21) Lines 149-150: I do not believe this sentence is saying what it is intended to say. I think this is saying that there were two cases that were detected in culture but not by tNGS, but the number of cases detected by tNGS is still higher than by culture. However, the way it is worded makes it seem like tNGS detected two fewer TB cases when compared to culture, which is not true. Please revise for clarity.

Response:

Thank you for highlighting the confusion. We have revised the sentence to clarify that although tNGS missed two culture-positive cases, these same cases were also negative by all other molecular tests. This suggests that the discrepancy may be due to low pathogen load or sample quality, rather than a limitation specific to tNGS.

(See revised **Lines 178-179**)

22) Lines 154-156: Are any of these differences statistically significant?

Response: Thank you for your thoughtful question. We agree that statistical testing of agreement metrics such as PPA, NPA, and OPA would provide additional context. However, given the limited number of positive and negative samples in each comparison group—particularly for extrapulmonary TB (EPTB)—we did not perform formal statistical tests (e.g., McNemar’s test or confidence intervals for differences in proportions), as the small sample sizes would limit statistical power and increase the risk of type II error.

Instead, our aim was to provide descriptive consistency metrics (PPA, NPA, OPA) to highlight the agreement trends between tNGS and the reference molecular tests (Xpert and TB-DNA). For example:

(1) In all samples, the OPA between tNGS and Xpert was 86.8%, and between tNGS and TB-DNA was 85.2%, indicating similar overall agreement.

(2) In PTB samples, these trends were consistent, with an OPA of 87.3% and 85.5%, respectively.

(3) In EPTB samples, the numbers were too small (e.g., n=13 and n= 5) to draw reliable statistical conclusions.

We have now clarified this limitation in the manuscript (Lines 303–308) and added a statement to explain that formal statistical comparisons were not conducted due to the small subgroup sizes.

23) Lines 163-164: How was this cohort chosen?

Response:

Thank you for raising this question. We have now clarified the inclusion and exclusion criteria for this cohort. Specifically, this subgroup included 85 BALF and 1 lung tissue sample obtained via invasive procedures (e.g., bronchoscopy or biopsy) from patients who were clinically diagnosed, suspected, or confirmed to have TB. Only high-quality respiratory tract samples with sufficient volume and corresponding clinical diagnosis were included. Classification into “confirmed TB” and “clinically diagnosed/suspected TB” was based on a composite diagnostic standard involving microbiological evidence and evaluation by at least three experienced TB clinicians.

24) Throughout: Please spell out numbers less than ten (one instead of 1).

Response:

Thank you for the reminder. We have carefully reviewed the manuscript and revised all non-technical numbers less than ten to their spelled-out forms (**Changes has made throughout the manuscript**)

25) Lines 163-164: I don't necessarily agree with only looking at sensitivity and specificity within the TB positive patients. This could be used in comparison with total sensitivity and specificity, but the intent of the assay is to detect TB in ANY patient.

Response:

Thank you for the thoughtful comment. We agree that the primary purpose of diagnostic assays is to detect TB in all patients, and we have accordingly reported the overall sensitivity and specificity of tNGS versus other methods in the main Results section. The paragraph in question was intended to illustrate performance within specific subgroups, particularly the “confirmed TB” cohort, to compare positivity rates of different methods under a more controlled diagnostic reference. To address potential confusion, we have revised the wording to clearly indicate that this analysis reflects subgroup positivity rather than overall diagnostic sensitivity. (See revised

Lines 191-196,197-200)

26) Lines 169-170: Why are the denominators different (17 vs 16)?

Response:

Thank you for the observation. The difference in denominators is due to incomplete testing in a small number of patients. As this is a retrospective study, not all patients underwent every diagnostic test, which is a known limitation of real-world clinical datasets. We have clarified this point in the revised text.

27) Line 184: 40% sensitivity should not be described as "high sensitivity."

Response:

Thank you for pointing this out. We agree that 40% sensitivity should not be described as “high.” We have revised the sentence to remove this phrase and present the data more objectively. (See **Line 214-216)**

28) Lines 186-187: Extrapulmonary TB specimen sources would be considered "off-label" use of the assay. This should be included in the discussion.

Response:

Thank you for this insightful comment. We have clarified in the Methods section that the GeneXpert MTB/RIF assay is officially validated for use with respiratory

specimens, while extrapulmonary samples are considered off-label applications. We have also added a statement in the Discussion noting that, in contrast, tNGS is not limited by sample type and can be applied to both pulmonary and extrapulmonary specimens, thus expanding its diagnostic applicability in clinical settings. (See **Lines 128-132,263-267**)

29) Line 191: "Xpert and tNGS" instead of "Xpert or tNGS?"

Response:

Thank you for your question. In this analysis, we specifically focused on samples that tested positive for *M. tuberculosis* by either Xpert or tNGS, as shown in Figure 2A. This approach ensured that we evaluated resistance detection only in samples where the pathogen was actually detected by at least one molecular method. We believe this is an appropriate precondition for downstream resistance interpretation and have clarified this rationale in the revised text.

(See revised **Line 221-222**)

30) Line 202: Define "confusion matrix."

Response:

Thank you for your suggestion. We have added a brief definition of the “confusion matrix” in the revised text, clarifying that it refers to a 2×2 table used to compare test results against the reference standard in terms of true positives, false positives, false negatives, and true negatives.

(See revised **Line 233-234**)

31) Lines 203-205: This sentence needs to be revised.

Response:

Thank you for your suggestion. We have revised the sentence for improved clarity and flow. The reference to Table S5 has been integrated appropriately, and the detection rates of specific NTM subtypes are now clearly stated.

(See revised **Lines 228-231**)

32) Line 204: *Mycobacterium avium* should be italicized.

Response:

Thank you for pointing this out. We have corrected the formatting, and *Mycobacterium avium* now appears in italics, as per taxonomic conventions.

(See revised **Line 235**)

33) Line 208: "However" can be removed from the beginning of this sentence.

Response:

Thank you for the suggestion. We have removed “However” from the beginning of the sentence to improve clarity and flow.

(See revised **Line 240**)

34) Line 210: "Following this," can be removed from the beginning of this sentence.

Response:

Thank you for your suggestion. We have removed “Following this,” from the beginning of the sentence to improve conciseness and readability.

(See revised **Line 242**)

35) Line 224: "Etc." instead of "et al."?

Response:

Thank you for catching this. We agree that “etc.” is more appropriate in this context, as the sentence refers to an incomplete list of items rather than a reference citation.

We have made the correction accordingly.

(See revised **Line 262**)

36) Line 240: Please describe how detection of other pathogens has a "substantial clinical significance."

Response:

Thank you for the opportunity to clarify this point. We have expanded the results section to explain that detecting co-infecting pathogens may assist in tailoring empirical antimicrobial therapy, identifying patients at risk of co-infections, and avoiding unnecessary anti-TB treatment in non-TB cases. This is particularly relevant in settings where TB-like symptoms may be caused or complicated by other bacterial, fungal, or viral pathogens.

(See revised **Line 245-251**)

37) Line 247: How does tNGS enhance the understanding of TB's interaction with other infectious agents? Please clarify.

Response:

Thank you for the helpful suggestion. We have revised the sentence to clarify that tNGS enables simultaneous detection of *M. tuberculosis* and co-infecting pathogens, which allows for the identification of co-infection patterns. This can provide insight into pathogen–pathogen interactions, help explain atypical clinical presentations, and inform tailored treatment decisions.

(See revised **Line 292-295**)

38) Table 1: For the p-value, what is being compared to what?

Response:

Thank you for the question. The p-values in Table 1 reflect comparisons of baseline clinical and demographic characteristics among the three diagnostic groups: pulmonary TB (PTB), extrapulmonary TB (EPTB), and non-TB (NTB). Categorical variables were compared using chi-square tests. Continuous variables were compared using the Kruskal–Wallis test. We have clarified the statistical methods used in the revised table legend. (See updated **Table 1 legend**)

39) Figure 1C: "Clinically" instead of "Clinical."

Response:

Thank you for pointing this out. We have corrected the label in Figure 1C from “Clinical” to “Clinically” to ensure proper grammar and clarity.

(See **Figure 1C**)

40) Figure 3: Please list how the other pathogens were detected? tNGS? Culture? Multiple methods?

Response:

Thank you for the comment. All pathogens displayed in Figure 3 were detected by targeted next-generation sequencing (tNGS) only. We have updated the figure legend to clarify this. (See Figure 3 Legend)

41) Figure 3: Would prefer actual numbers instead of a detected ratio.

Response:

Thank you for the suggestion. We have revised Figure 3 to display the absolute number of detections (n) for each pathogen instead of detection ratios. This change improves clarity and clinical interpretability. The figure legend has also been updated accordingly. (See revised **Figure 3**)

42) Figure 3: Why is *Mycoplasma* separated out from other bacteria?

Response:

Thank you for the thoughtful question. In our clinical diagnostic practice, *Mycoplasma* species are typically treated as a distinct category of pathogens due to their unique biological and clinical characteristics—such as lack of a cell wall, distinct antibiotic susceptibility patterns, and their relevance in atypical pneumonia. For this reason, we displayed *Mycoplasma* separately from other bacterial pathogens in Figure 3 to align with common clinical interpretation and reporting conventions.

43) Supplemental methods: What 64 species were included in this panel? Can you include this as a table? See Major Comment 1.

Response:

Thank you for this important comment. We have now included the full list of 64 targeted pathogens in **Supplementary Table 6**, which outlines all bacterial, fungal, viral, and atypical organisms covered by the tNGS panel used in this study.

(See new **Supplementary Table 6**)

44) Supplemental Table 1: It would be helpful to know the positivity rate for each sample type. Could you include this in the table?

Response:

Thank you for the helpful suggestion. We have updated **Supplementary Table 1** to include the positivity rate of *M. tuberculosis* detection for each sample type. The positivity rate was calculated as the number of positive cases divided by the total number of tested samples per specimen type. (See revised **Supplementary Table 1**)

Reviewer #2 (Comments for the Author):

Overview: Dai et al. addresses a relevant public health issue, TB, and use of targeted

next-generation sequencing (tNGS). Comparative analysis with conventional methods and Xpert MTB/RIF is scientifically sound. As well as reporting of sensitivity percentages for different sample types adds value.

Minor:

Line 28 and 29 - Say effectiveness twice under the same context - can be combined.

Response:

Thank you for your careful reading. This issue was also noted by Reviewer #1. We have revised the sentence in Lines 28–29 to eliminate the redundancy and improve clarity. (See **Lines 28-29**)

Line 51 - Difficult to diagnose extrapulmonary specimens - is this accurate? It's the suspicion of extra pulmonary TB that is challenging not necessarily the sensitivity and specificity of testing in extrapulmonary sites.

Response:

Thank you for this insightful comment. We fully agree that the main challenge in extrapulmonary TB (EPTB) lies in the difficulty of early clinical suspicion due to its often non-specific or atypical presentation, rather than in the performance of diagnostic tests themselves. We have revised the sentence accordingly to clarify this point. (See revised **Line 47-49**)

Line 84-85 - TB-DNA, T-SPOT and QFT need to be defined prior to using short forms.

Response:

Thank you for your suggestion. We have revised the text to define TB-DNA, T-SPOT.TB, and QFT (QuantiFERON-TB Gold) assays at their first mention in the Methods section, including assay principles and manufacturers. (See revised **Lines 99-101**)

Line 103 - Define what the MPT64 antigen detection method is

Response:

Thank you for your suggestion. We have revised the sentence to clarify that the MPT64 antigen detection method is an immunochromatographic assay that identifies the MPT64 protein secreted by members of the *Mycobacterium tuberculosis* complex.

(See revised **Lines 122-124**)

Line 105-107 - May be worth stating or at least referencing that the Xpert uses the 81-bp RRDR region for determining resistance and even though it can catch most but not all Rifampicin resistant TB.

Response:

Thank you for the insightful comment. We have added a sentence to clarify that the Xpert MTB/RIF assay detects mutations within the 81-bp rifampicin resistance-determining region (RRDR) of the *rpoB* gene. As highlighted in the WHO mutation catalogue, while this region includes the majority of known rifampicin resistance-associated mutations, certain rare mutations outside this region may not be detected.

(See revised **Lines 128-130**)

Line 134 - PTB and EPTB were not defined previously prior to using

Response:

Thank you for your suggestion. We have now added the definitions of PTB (pulmonary tuberculosis) and EPTB (extrapulmonary tuberculosis) at their first mention in the Introduction section to ensure clarity for readers.

(See revised **Lines 55-56**)

Line 135 - What diagnostic guidelines?

Response:

Thank you for your comment. We have clarified the diagnostic criteria used in the study. Specifically, grouping was based on clinical judgment according to the World Health Organization (WHO) guidelines and the Chinese national diagnostic standard for pulmonary tuberculosis (WS 288–2017). This clarification has now been added to the revised manuscript (see Line **82-94**).

Line 142 - 144 - Either in the table or within the paper you may want to define WBC%, NEU% etc.

Response:

Thank you for pointing this out. We have now included the full definitions of WBC (white blood cell count), NEU% (neutrophil percentage), and other related clinical indicators at their first mention in the main text, and have also added a comprehensive list of abbreviations in the footnote of Table 1 for clarity.(See **Table1**)

Line 161 - Extrapulmonary does not need the capital E error and changed “Extrapulmonary” to “extrapulmonary” in Line 161.

Response:

Thank you for your careful review. We have corrected the capitalization. (See **Line 190**)

Line 191 - seems to be an error "with at least one sample testing positive for TB"

Response:

Thank you for pointing this out. We agree that the original sentence was unclear and may have caused confusion. We have revised the sentence.
(See **Lines 221-222**)

Major:

Line 163 - 85 BALF and 1 tissue sample are these all from unique patients.

Response:

Thank you for this important clarification. Yes, each of the 85 BALF samples and the one lung tissue sample was obtained from a unique patient. We have revised the sentence.

Line 163 - 168 - Inconsistent denominators for all of the testing. The variable sample set is a limitation of retrospective studies, however it is challenging to truly gather sensitivity or specificity in such a manner. It may make sense to normalize to one test group - all positives by any method vs tNGS

Response:

Thank you for this insightful comment. We acknowledge that the denominators varied across methods due to the retrospective nature of the study, in which not all patients underwent every diagnostic test. This variability indeed represents a limitation. To address this, we did not use any single method (e.g., culture or Xpert) as the reference but rather employed a composite clinical reference standard—incorporating microbiological, radiological, and clinical findings evaluated by a panel of experienced TB clinicians—to classify patients into confirmed, clinically diagnosed, and suspected TB categories. This approach allowed us to evaluate the clinical diagnostic performance of tNGS in a real-world context despite the heterogeneity in testing.

We have added clarifications to both the Methods and Discussion sections to highlight this design and its implications. Thank you again for the helpful suggestion. (See Lines 304-309)

Line 164 - What is the confirmed TB category - how is this defined?

Response:

Thank you for the comment. We have added the specific diagnostic criteria for *confirmed*, *clinically diagnosed*, and *suspected* TB groups to the Methods section to clarify the basis of classification. (See **Lines 82-94**)

Line 167 - 168 - what is the correlate for saying tNGS detected 7 more TB cases?

How can you be sure they are not false positives given all other tests were negative?

Response:

Thank you for raising this important point. The 7 additional TB-positive cases detected by tNGS were not confirmed by conventional microbiological tests; however, all of them had consistent clinical presentations, radiological findings, and a final discharge diagnosis of TB. Furthermore, during one-month follow-up, these patients showed clinical improvement after anti-TB treatment, further supporting the likelihood of true-positive results rather than false positives.

Line 163 -168 - Insufficient statistical support given the lack of p-values and PPV or NPV.

Response:

We have added p-values and PPV in supplemental tables. (See table S3)

Line 169-173 - Similar issues with lack of consistency in denominators with a limited sample size. Unclear what defines "4 extra positives"

Response:

We appreciate the reviewer's insightful comment. As a retrospective study, sample-level variations in test availability (e.g., culture, Xpert, TB-DNA) resulted in inconsistent denominators, which we now explicitly acknowledge as a limitation in the Discussion section.

The phrase "4 extra positives" referred to sputum samples in the clinically diagnosed and suspected TB groups that tested positive only by tNGS but were negative by all conventional tests (AFB smear, culture, TB-DNA, and Xpert). These patients were classified as TB based on comprehensive clinical diagnosis, including compatible symptoms, imaging findings, discharge diagnosis, and documented clinical response to anti-TB treatment at one-month follow-up. We have now revised the main text to clarify this definition and added relevant clinical correlation to support these findings. (See Lines 82-94)

Line 174-177 - Similar issues with lack of consistency in denominators and how "extra positives" were confirmed.

Line 183 - 184 - Verify accuracy calculation of 68.3% - it is a bit odd given the high specificity and moderate sensitivity

Response:

We appreciate the reviewer's observation. The overall accuracy of 68.3% was calculated as $(TP + TN)/Total$. This relatively moderate value, despite high specificity, is primarily due to the imbalance in the number of TB-positive and TB-negative samples included in the study—specifically, a smaller number of TB-negative cases in the final cohort. As a result, true negative cases (which contribute to specificity) have a limited impact on the overall accuracy calculation. We have added a clarification of this limitation in the discussion section.

Line 186 - 188 - Was the Xpert validated for the extrapulmonary sites?

Response:

We agree with the reviewer that Xpert MTB/RIF has not been officially validated for all extrapulmonary specimen types. While it is widely used off-label for such samples in clinical settings, its sensitivity and specificity can vary significantly depending on the sample type. In our study, we included extrapulmonary samples based on real-world clinical testing practices and have added a statement in the Discussion to acknowledge the off-label nature of Xpert for these specimens.

Line 190-198 - Given that Xpert can only detect Rifampicin resistance, it would be helpful to know whether the additional detected resistance to Isoniazid and MDR and XDR is consistent with local epidemiology.

Response:

Thank you for this important question. To address this point, we reviewed the clinical and microbiological data of all patients in whom tNGS detected resistance mutations, particularly for isoniazid, ethambutol, streptomycin, fluoroquinolones, and second-line injectable drugs.

Among the 20 patients with tNGS-detected drug resistance, 6 cases showed high consistency with phenotypic drug susceptibility testing (pDST), and 3 cases were consistent with other molecular tests (e.g., Xpert). Notably, 8 of the patients were culture-negative and did not undergo pDST. In these cases, the comprehensive resistance profiling provided by tNGS enabled clinicians to adjust treatment regimens, leading to symptom improvement and positive clinical outcomes during follow-up.

The observed resistance patterns—including rates of MDR-TB and XDR-TB—are generally in line with regional epidemiological data from southwestern China.

Importantly, the resistance mutations identified by tNGS, particularly for core anti-TB drugs such as rifampicin (e.g., *rpoB* S450L, D435V), isoniazid (*katG* S315T/N), and fluoroquinolones (*gyrA* D94G/A), were directly used to inform clinical treatment decisions in many of these cases.

We have added this analysis and a brief summary of clinical correlation in the revised manuscript (Discussion, **Lines 278-285**).

Line 199-213 - Were the co-detection of other pathogens in these cases clinically relevant?

Response:

Thank you for your thoughtful comment. We agree that the clinical relevance of co-detected pathogens is important in evaluating the diagnostic value of tNGS. In the revised manuscript, we have provided specific clinical examples where the co-detection of additional pathogens was indeed clinically meaningful and influenced treatment decisions.

Among the tNGS-positive cases, one case revealed *Aspergillus fumigatus* co-infection via bronchoalveolar lavage fluid. This finding was consistent with radiological and clinical suspicion of invasive pulmonary aspergillosis (IPA), and antifungal therapy was initiated promptly, resulting in improvement of respiratory symptoms. In another case, *Nocardia cyriacigeorgica* was co-detected in a patient with cavitary lung lesions, leading to targeted treatment with sulfamethoxazole, which was associated with radiographic resolution.

Additionally, three other patients had been clinically suspected of fungal co-infection before tNGS, and the tNGS results confirmed the presence of fungal pathogens (e.g., *Candida albicans*), supporting the initiation or continuation of antifungal therapy. These examples support that tNGS not only identifies *Mycobacterium tuberculosis* but also provides actionable insights on co-infections, which may otherwise be missed by conventional methods. We have added a brief summary of these findings in the revised results section (See **Lines 245–251**).

Line 249 - Limitations of tNGS were listed here but not the limitations of the paper regarding a retrospective study, inconsistent testing across specimens, limited sample size etc.

Response:

We thank the reviewer for pointing out this important issue. We have now added a discussion on the limitations of our study design, including the retrospective nature of the study, inconsistent availability of testing results across different sample types due to real-world clinical workflows, and the relatively limited sample size, which may affect statistical power. These have been clearly acknowledged in the revised Limitations section (Lines **296–312**).

At the same time, we would like to emphasize that this real-world, retrospective cohort reflects current diagnostic practices in TB-endemic settings, and therefore provides practical insights into how tNGS performs under typical clinical constraints. This adds translational relevance to our findings.

Overall, the paper doesn't really discuss the limitations of tNGS in terms of the high cost for the assay which is a large limiting factor.

Response:

We agree that the current cost of tNGS remains a major challenge for implementation in resource-limited settings. We have now included this point in the Discussion (**Lines 296–300**), noting that while cost is a barrier, ongoing advancements in multiplex PCR, streamlined library prep, and sequencing technologies are expected to reduce costs in the near future. Furthermore, by consolidating pathogen detection and drug resistance profiling in one assay, tNGS may eventually improve cost-efficiency compared to performing multiple separate tests.

Sensitivity issues with detecting low organism burden which is really why culture really remains the gold standard.

Response:

Thank you for this valuable observation. We agree that culture remains essential, especially in paucibacillary cases. We have updated the manuscript (**Lines 300–304**) to reflect this and discuss that tNGS, while sensitive in moderate-to-high burden samples, may underperform in extremely low-burden cases. However, tNGS has demonstrated superiority over smear and comparable performance to culture in many

sample types — while also providing drug resistance and co-infection data from the same specimen, which culture cannot easily do.

Or it's limitation with turnaround time which is the advantage of assays like Xpert. It also doesn't discuss limitations of pre-analytical factors like the specimen quality etc.

Response:

We acknowledge that the current turnaround time of tNGS (typically 13 hours) is longer than that of rapid molecular tests such as Xpert. This has been added to the revised Limitations section (**Lines 300–304**). However, tNGS provides broader pathogen detection and comprehensive resistance information beyond rifampicin, which offers clinical value in complex or atypical cases where rapid tests are insufficient.

Additionally, that it is a single center retrospective study that may limit generalizability of the findings.

Response:

We fully agree and have now included this as a key limitation in the Discussion section (**Lines 304–309**). While this is a limitation, it also allowed for consistency in clinical management, diagnostic workflows, and sample handling, which minimizes inter-center heterogeneity and improves internal validity.

The paper also doesn't discuss the bioinformatic tools and evaluation needed for determining resistance.

Response:

Thank you for this suggestion. We have added a description of the bioinformatic pipeline used for variant calling and resistance annotation, including reference databases and mutation curation (See **Supplement methods**). This addition enhances the transparency and reproducibility of our resistance detection approach

Claims in this study in terms of tNGS sensitivity does not seem sufficient and are overstated without consistent testing.

Response:

We appreciate the reviewer's caution and have revised the relevant sections to avoid overgeneralization. Sensitivity and specificity results are now consistently reported with denominators specified for each comparator method. While inconsistent testing is a limitation of the retrospective design, our results still demonstrate that tNGS achieved robust performance across a range of clinical samples, particularly when compared to traditional methods in real-world settings. These findings support the potential of tNGS as a valuable complementary tool, especially in complex or extrapulmonary TB cases.

Re: Spectrum01698-25R1 (**Targeted Next-Generation Sequencing for Comprehensive Diagnosis and Drug Resistance Detection in Pulmonary and Extrapulmonary Tuberculosis: A Single-Center Retrospective Study**)

Dear Dr. Guiming Liu:

Thank you again for revising and re-submitting your work. While the manuscript is much improved from the initial submission, there were additional major comments that need to be addressed. Below you will find my comments, instructions from the Spectrum editorial office, and the reviewer comments.

Revision Guidelines

Sincerely,
Kenneth Gavina
Editor
Microbiology Spectrum

Reviewer #1 (Comments for the Author):

This is a revised manuscript that was previously reviewed, and I would like to thank the authors for the time they spent responding to the comments. The inclusion of Supplemental Table 6 is very helpful and important for the interpretation of the results presented in this manuscript.

Major Comments:

- 1) Lines 160-161: What does this sentence mean? Which samples were used for the study? Only the first sample submitted from an individual patient or any sample from any patient, as available? Please clarify.
- 2) Lines 223-231: It says that Xpert detected six cases of rifampin resistance while tNGS detected one less (five cases). However, at the end of this paragraph, it says there are seven cases of rifampin resistant TB. Please clarify the number of isolates that were rifampin resistant and ensure consistency throughout.
- 3) Lines 226-227: This sentence is confusing because it reads like there were 22 resistance mutations detected across the 20 samples, but it's really saying that 20 samples had at least one of the 22 resistance mutations tested by the assay. Please update for clarity.
- 4) Lines 247-254: This new section is interesting, but there isn't a discussion about clinical impact or outcomes anywhere else in the manuscript. While I do think it is very important to document the true clinical impact of using tNGS vs other diagnostic assays, that's not really the purpose of this manuscript and should either be done for all samples or no samples. This could potentially be moved to the discussion or it needs to be a more significant aspect of this paper.
- 5) Lines 283-285: Was clinical effectiveness proven in all eight cases, or only some of the cases? Additionally, this is a bit misleading because treatment success was not a metric used to establish the utility of tNGS in the manuscript. Were the cases identified by Xpert or TB DNA successfully treated, too? See Major Comment 4.
- 6) Lines 297-299: There is no supporting evidence within this manuscript to suggest that tNGS can help elucidate pathogen-pathogen interactions. The assay can simply detect MTBC and select group of other organisms with pathogenic potential.

Minor Comments:

- 1) Lines 33-34: Pulmonary TB (PTB) and extrapulmonary TB (EPTB) instead. EPTB is misspelled.
- 2) Line 34: Define BALF.
- 3) Lines 40-42: The use of tNGS for identification of other pathogens or drug detection is not listed in the background of the abstract and just shows up in the conclusions. This should be included in the background section.
- 4) Throughout: Organism names should be italicized.
- 5) Lines 52-54: tNGS is still relatively expensive and resource intensive. How will this help patients in underserved settings?
- 6) Line 61: Define AFB.
- 7) Lines 80-95: Can you please clarify how samples were processed for both routine microbiology and tNGS? Were samples shared or were residual samples from routine testing used for tNGS.
- 8) Lines 170-171: If using the word "significantly" there needs to be a statistical analysis to show true significance.
- 9) Lines 172-173: Consider using semicolons between parameters to better clarify which acronym goes with which test.
- 10) Lines 173-174: See Minor Comment 8 for using the word "significant" here.
- 11) Lines 177-183: Please include the total number of negative cases.
- 12) Line 201: "Six" instead of 6.
- 13) Lines 201-202: "Notably" is used to start both sentences. Please update for better readability.
- 14) Line 207: How were the seven TB cases confirmed?
- 15) Lines 209-210: Can you please clarify how many samples are in the clinically diagnosed EPTB group, and how many were detected by each method? The word additional is confusing.
- 16) Lines 216-218: The wording here is a bit weird. Please update for clarity.
- 17) Lines 223-224: How were the 30 samples chosen? Was this randomized in some way? Please clarify.
- 18) Line 229: "Nine" instead of 9.
- 19) Line 230: "Three" instead of 3.
- 20) Line 241: EBV and CMV do not need to be italicized.
- 21) Line 245: Please update to "seven of the 13 samples (54%)" for consistency.
- 22) Lines 246-247: This sentence is a little confusing as written. Is this saying that there was one case of PTB that had MTBC and NTM and one case of EPTB that had MTBC and NTM detected?
- 23) Line 284: Define DST.
- 24) All figures/tables: Define all acronyms used in the figure/table in the figure/table legend.
- 25) Supplemental methods (tNGS assay): Consider "custom" instead of "specially designed."
- 26) Supplemental Figure 1: Numbers don't match up in the pulmonary TB->clinically diagnosed TB (n=32 and then n=33).
- 27) Supplemental Figure 1: So, NTM weren't detected in ANY non-TB cases? There were only two cases where an NTM was detected, and it was detected with MTBC (See Minor Comment 22)? What is the purpose of showing the NTM arm in the figure? Only culture can detect NTM, so unless there is an NTM causing a false positive result, all of these should be negative.
- 28) Supplemental Table 5: Total=21 (not 20).
- 29) Supplemental Table 6: Please add that HHV-4 is EBV and that HHV-5 is CMV because those are how they are listed in the manuscript.
- 30) Supplemental Table 6: Pneumocystis jirovecii is misspelled.

Reviewer #2 (Comments for the Author):

- There are several redundancies within line 155-156 could be made more succinct and clear
- The results section especially the first three paragraphs include a lot of methodology that can be re-written to include within methods such that the results section can solely focus on conclusions and start off with Baseline clinical characteristics.
- It would significantly help the results section and the reader come to the same conclusions, if the author explicitly clarified

which two assays are being compared and which one performed significantly better for eg. tNGS vs Xpert.

Re-review:

Targeted Next-Generation Sequencing for Comprehensive Diagnosis and Drug Resistance Detection in 2 Pulmonary and Extrapulmonary Tuberculosis: A Single-Center Retrospective Study

- There are several redundancies within line 155-156 could be made more succinct and clear
- The results section especially the first three paragraphs include a lot of methodology that can be re-written to include within methods such that the results section can solely focus on conclusions and start off with Baseline clinical characteristics.
- It would significantly help the results section and the reader come to the same conclusions, if the author explicitly clarified which two assays are being compared and which one performed significantly better for eg. tNGS vs Xpert.

Dear Editor,

We sincerely thank you and the reviewers for the thoughtful and constructive comments on our manuscript "**Targeted Next-Generation Sequencing for Comprehensive Diagnosis and Drug Resistance Detection in Pulmonary and Extrapulmonary Tuberculosis: A Single-Center Retrospective Study**" (Manuscript ID: Spectrum01698-25). We have carefully revised the manuscript in accordance with all suggestions. All comments have been addressed point by point in the following **Response to Reviewers**, and the corresponding changes are marked in the revised manuscript. We believe these revisions have improved the clarity, structure, and scientific rigor of the work, and we respectfully resubmit the manuscript for your consideration.

With kind regards,

Guiming Liu

(on behalf of all authors)

Response to Reviewers

Reviewer #1 (Comments for the Author):

This is a revised manuscript that was previously reviewed, and I would like to thank the authors for the time they spent responding to the comments. The inclusion of Supplemental Table 6 is very helpful and important for the interpretation of the results presented in this manuscript.

Major Comments:

1) Lines 160-161: What does this sentence mean? Which samples were used for the study? Only the first sample submitted from an individual patient or any sample from any patient, as available? Please clarify.

Response:

Thank you for your comment. The intention of the original sentence was to clarify that each patient was represented by one clinical specimen in the analysis and that all microbiological assays and tNGS were performed on the same specimen to enable a direct, sample-level comparison.

This was a retrospective real-world study based on routine diagnostic practice. We did not prospectively control the order or number of specimens submitted by each patient. For the purposes of this study, each enrolled patient contributed a single specimen—the clinical specimen that had been processed for tNGS as part of routine TB

diagnostics. In cases where more than one specimen had been collected from the same patient in routine care, only the specimen on which tNGS was performed and for which complete microbiological results were available was included in the analysis; other specimens from the same patient were not analyzed. Therefore, the samples used in this study were not necessarily always the first specimens submitted but were consistently the same specimens on which both tNGS and the comparator microbiological tests had been performed.

We agree that the previous wording in the Results section was confusing and not appropriately placed. In the revised manuscript, we have removed this sentence from the Results and rewritten and relocated the description to the Methods (**Lines 85-94**) to more clearly describe the retrospective enrollment and sample handling.

2) Lines 223-231: It says that Xpert detected six cases of rifampin resistance while tNGS detected one less (five cases). However, at the end of this paragraph, it says there are seven cases of rifampin resistant TB. Please clarify the number of isolates that were rifampin resistant and ensure consistency throughout.

Response:

Thank you for this helpful comment. We agree that our original wording mixed two different denominators in the same paragraph and may have been confusing.

For the head-to-head comparison between tNGS and Xpert, we restricted the analysis to 30 TB-positive specimens that (a) were positive by either Xpert or tNGS and (b) had valid results from both assays (Figure 2A). Within this 30-specimen subset, Xpert reported rifampicin resistance in six cases, whereas tNGS detected *rpoB* resistance mutations in five of these six cases, yielding a positive percent agreement of 83.3% (5/6). In the same 30 specimens, tNGS additionally identified resistance mutations to drugs other than rifampicin (e.g., isoniazid, fluoroquinolones) in seven patients, which are not captured by Xpert, as it only reports rifampicin resistance.

When we then extended the analysis to all tNGS-positive TB specimens in the cohort (beyond the 30-specimen concordance subset), tNGS detected a total of 22 resistance-conferring mutations across 20 patients (Figure 2B). Among these, *rpoB* S450L, associated with rifampicin resistance, was observed 12 times, and *katG* S315N, associated with isoniazid resistance, was detected in 9 instances. Because some specimens harbored more than one resistance mutation and clinical classification was performed at the patient level according to WHO definitions, these mutation patterns corresponded to seven patients with

rifampicin-resistant TB (RR-TB), of whom three fulfilled criteria for multidrug-resistant TB (MDR-TB) and two for pre-extensively drug-resistant TB (pre-XDR-TB). These seven RR-TB patients were identified in the full tNGS cohort and therefore are not limited to the 30 specimens used for the tNGS–Xpert concordance analysis. The difference in denominators explains why the number of rifampicin-resistant cases in the Xpert–tNGS concordance subset (6 Xpert-RIF-R, 5 tNGS-RIF-R) differs from the total of seven RR-TB patients reported for the entire tNGS-positive cohort.

To improve clarity, we have rewritten this paragraph in the Results to more clearly separate (a) the 30-specimen concordance analysis from (b) the resistance spectrum in the full cohort, and we explicitly state the denominators in each part. See the revised text in **Lines 219–224**.

3) Lines 226-227: This sentence is confusing because it reads like there were 22 resistance mutations detected across the 20 samples, but it's really saying that 20 samples had at least one of the 22 resistance mutations tested by the assay. Please update for clarity.

Response:

Thank you for this helpful clarification. Our intention was to indicate that, in the full tNGS-positive cohort, 20 patients carried at least one of the 22

distinct resistance-conferring mutations covered by the assay, rather than that exactly 22 mutation events were observed.

As this issue is closely related to Major Comment 2, we have now aligned the wording and made the meaning more explicit. The corresponding sentence in the Results has been revised (**Lines 224–229**).

4) Lines 247-254: This new section is interesting, but there isn't a discussion about clinical impact or outcomes anywhere else in the manuscript. While I do think it is very important to document the true clinical impact of using tNGS vs other diagnostic assays, that's not really the purpose of this manuscript and should either be done for all samples or no samples. This could potentially be moved to the discussion or it needs to be a more significant aspect of this paper.

Response:

Thank you for this helpful comment. We agree that the primary objective of this study is to evaluate the diagnostic performance of tNGS, and that our dataset does not support a systematic, cohort-wide analysis of clinical outcomes associated with tNGS-guided management. The cases originally described in Lines 247–254 were intended only as illustrative examples of potential clinical utility rather than as a rigorous outcome analysis.

In line with your suggestion, we have removed this paragraph from the Results section and incorporated it into the existing Discussion on co-detected pathogens, so that these cases are presented as illustrative examples rather than as formal outcome analyses. In the revised text, these cases are used to highlight the potential of tNGS for detecting clinically relevant co-infections, while the wording has been adjusted to avoid over-confident claims and to clearly indicate that they derive from a small, non-systematically ascertained subset of patients (See **Lines 281–289**) .

5) Lines 283–285: Was clinical effectiveness proven in all eight cases, or only some of the cases? Additionally, this is a bit misleading because treatment success was not a metric used to establish the utility of tNGS in the manuscript. Were the cases identified by Xpert or TB DNA successfully treated, too? See Major Comment 4.

Response:

Thank you for raising this important point. The sentence referring to “eight culture-negative cases where no phenotypic DST was available” was added in a previous revision in response to earlier reviewer concerns about the value of resistance calls in the absence of pDST or direct comparison with Xpert. Our intention was to illustrate how clinicians sometimes used tNGS resistance results to adjust anti-TB regimens in

culture-negative cases. However, as you correctly note, treatment success was not prospectively defined as an endpoint, and no rigorous, pre-planned evaluation of clinical outcomes was performed for these cases.

Given that the design and data of the present study only support a descriptive assessment of resistance mutations detected by tNGS, and not a formal evaluation of tNGS-guided treatment effectiveness, we have removed this sentence from the Results section. The corresponding paragraph now focuses solely on the resistance mutations detected by tNGS and the resulting classification of RR-TB, MDR-TB, and pre-XDR-TB, without invoking treatment response. As the analytical and clinical performance of tNGS for detecting resistance mutations has already been established in several dedicated studies, the present work is confined to reporting the resistance mutations observed in our cohort rather than re-evaluating clinical effectiveness.

6) Lines 297-299: There is no supporting evidence within this manuscript to suggest that tNGS can help elucidate pathogen-pathogen interactions. The assay can simply detect MTBC and select group of other organisms with pathogenic potential.

Response:

Thank you for this clarification. We agree that, given the design of our study and the limited number of illustrative cases, our data do not

support claims that tNGS can elucidate pathogen–pathogen interactions in a mechanistic or causal sense.

As noted in our response to Major Comment 4, we have therefore revised the entire Discussion paragraph regarding the detection of co-infecting pathogens. Specifically, we have removed the original wording suggesting that tNGS can “help elucidate pathogen–pathogen interactions” and now restrict our conclusions to what is directly supported by the data—namely, that tNGS can detect *M. tuberculosis* and multiple other pathogens in a single assay and of revealing co-infection patterns at a descriptive level. We also explicitly state that only a small number of non-systematically ascertained cases are presented and that the study was not designed to formally evaluate interaction mechanisms or clinical outcome differences (See **Lines 289–294**) .

Minor Comments:

1) Lines 33-34: Pulmonary TB (PTB) and extrapulmonary TB (EPTB) instead. EPTB is misspelled.

Response:

Thank you for pointing this out. We have corrected the spelling of extrapulmonary TB and standardized the terms as “pulmonary TB (PTB)” and “extrapulmonary TB (EPTB)” at their first appearance in the Abstract and throughout the manuscript (see **Lines 34**).

2) Line 34: Define BALF.

Response:

Thank you for this comment. We have added the full term

“bronchoalveolar lavage fluid (BALF)” at its first occurrence in the manuscript to define the abbreviation (**see Lines 35**).

3) Lines 40-42: The use of tNGS for identification of other pathogens or drug detection is not listed in the background of the abstract and just shows up in the conclusions. This should be included in the background section.

Response:

Thank you for this helpful suggestion. We have revised the Background section of the Abstract to explicitly mention the use of tNGS for detecting drug-resistance mutations and co-infecting pathogens (**See Lines 28–29** in the revised abstract).

4) Throughout: Organism names should be italicized.

Response:

Thank you for pointing this out. We have carefully reviewed the manuscript and standardized the formatting of all organism names (e.g.,

Mycobacterium tuberculosis, *M. tuberculosis complex*, *Aspergillus*

fumigatus, *Nocardia cyriacigeorgica*, *Candida albicans*, etc.) to italics throughout the text, tables, and figure legends where applicable.

5) Lines 52-54: tNGS is still relatively expensive and resource intensive. How will this help patients in underserved settings?

By leveraging advanced sequencing technologies, our findings highlight a diagnostic approach that has the potential to improve TB diagnosis and support appropriate treatment strategies where tNGS is available, and that may increasingly benefit patients in underserved settings as sequencing platforms become more accessible and costs continue to decrease.

Response:

Thank you for this thoughtful comment. In the “Importance” paragraph, our aim was to provide a concise statement of the potential value of tNGS, while more detailed considerations regarding cost and implementation in resource-limited settings are discussed in the Limitations section. As noted there, tNGS is currently more expensive and dependent on sequencing infrastructure than conventional tests, but it can simultaneously detect *M. tuberculosis*, drug resistance mutations and co-infecting pathogens in a single assay, which may be particularly useful in selected scenarios such as suspected drug-resistant TB or complex mixed infections. In a previous work from the same

setting, the per-test costs were approximately 50 USD for culture, 70–200 USD for phenotypic DST, 110 USD for Xpert MTB/RIF, and 150 USD for tNGS, indicating that tNGS is more costly than individual conventional tests but can consolidate multiple diagnostic functions into a single procedure. With further dissemination of sequencing platforms and continued optimization of targeted NGS workflows, the cost of tNGS is expected to decrease, which may improve its feasibility in a broader range of settings over time.

To better reflect this view and directly address your concern, we have slightly revised the final sentence of the “Importance” paragraph (see **Lines 53–55**).

6) Line 61: Define AFB.

Response:

Thank you for this comment. We have added the full term “acid-fast bacilli (AFB)” at its first occurrence in the manuscript to define the abbreviation (see **Lines 62**).

7) Lines 80-95: Can you please clarify how samples were processed for both routine microbiology and tNGS? Were samples shared or were residual samples from routine testing used for tNGS.

Response:

Thank you for highlighting this point. As also clarified in our response to Major Comment 1, this was a retrospective real-world study in which we first identified clinical specimens that had undergone tNGS as part of routine TB diagnostics and then retrieved the corresponding routine microbiological results for the same specimens. For each enrolled patient, one respiratory or extrapulmonary specimen collected for routine TB workup was included, and all enrolled specimens underwent tNGS. We then retrospectively extracted results of culture, AFB smear, Xpert and TB-DNA performed on these same specimens from the medical record system. Thus, all comparative analyses were based on a single clinical specimen per patient on which both tNGS and the conventional microbiological tests had been performed, rather than on separate or residual samples (see **Lines 85–94**).

8) Lines 170-171: If using the word "significantly" there needs to be a statistical analysis to show true significance.

Response:

Thank you for this comment. Baseline clinical characteristics of the PTB, EPTB and NTB groups were statistically compared and are summarized with corresponding p-values in Table 1. In the revised manuscript, we have added the relevant p-values at the corresponding sentences in the Results and explicitly referred the reader to Table 1 for detailed statistics,

so that any use of “significantly” is directly supported by the underlying analysis (see **Lines 165**).

9) Lines 172-173: Consider using semicolons between parameters to better clarify which acronym goes with which test.

Response:

Thank you for this helpful suggestion. We have replaced the commas with semicolons between the parameters and their acronyms to improve readability and make the pairing clearer (see **Lines 168**).

10) Lines 173-174: See Minor Comment 8 for using the word "significant" here.

Response:

Thank you for this comment. We have added the p-values at this point to support the use of “significant” (see **Lines 168**).

Response:

11) Lines 177-183: Please include the total number of negative cases.

Response:

Thank you for this suggestion. We have updated the performance description to include the corresponding numerators and denominators, thereby explicitly conveying the total number of positive and negative

cases in the comparison with culture: "PPA 91.7% (22/24), NPA 73.9% (68/92), and OPA 77.6% (90/116)" (see **Lines 173–174**).

12) Line 201: "Six" instead of 6.

Response:

Thank you for pointing this out. We have replaced the numeral with the written form "Six" at this position (see **Lines 196**).

13) Lines 201-202: "Notably" is used to start both sentences. Please update for better readability.

Response:

Thank you for this suggestion. We have revised the wording to avoid repeating "Notably" and to improve readability. The sentences now read (see **Lines 196-197**):

"All six sputum samples tested with Xpert showed positive results, yielding a positivity rate of 100%. In addition, tNGS detected four TB-positive cases in the clinically diagnosed and suspected TB groups that were negative by all conventional methods (culture, AFB staining, TB-DNA, and Xpert)."

14) Line 207: How were the seven TB cases confirmed?

Response:

Thank you for this comment. In the extrapulmonary subgroup, “confirmed TB” was defined according to our composite reference standard as having at least one positive conventional microbiological test result, either on the same extrapulmonary specimen or on a follow-up specimen from the same disease episode.

Among the seven patients classified as confirmed TB in the 33 extrapulmonary samples, six had at least one positive conventional test on the same extrapulmonary specimen (one culture-positive, one TB-DNA-positive, two Xpert MTB/RIF-positive, and two AFB smear-positive).

The remaining case had negative results on all conventional tests performed on that extrapulmonary specimen, but the diagnosis of extrapulmonary TB was subsequently established based on pleural effusion findings and pathological confirmation. This patient had a history of tuberculous pleurisy and developed encapsulated pleural effusion after one year of anti-TB treatment. During the current hospitalization, pleural puncture drainage and subsequent surgical debridement were performed, and histopathological examination of the resected thickened pleura and necrotic tissue demonstrated features consistent with tuberculosis, thereby confirming the diagnosis according to the composite reference standard. We have also updated the description of the composite reference standard in the Methods to

explicitly include histopathological evidence as one of the criteria for confirmed TB (see **Line 100**).

15) Lines 209-210: Can you please clarify how many samples are in the clinically diagnosed EPTB group, and how many were detected by each method? The word additional is confusing.

Response:

Thank you for this helpful comment. In the extrapulmonary cohort, there were 19 specimens in the clinically diagnosed EPTB group. Among these, tNGS detected MTBC in 6/19 specimens, all of which were negative or untested by conventional microbiological methods (culture, AFB smear, Xpert and TB-DNA), as shown in Figure 1A. The remaining clinically diagnosed EPTB specimens were negative by all microbiological assays, including tNGS, and were classified as TB according to the composite clinical criteria described in the Methods.

To improve clarity and avoid the ambiguous term “additional,” we have revised the text to explicitly report these numbers (see **Lines 204–206**).

To facilitate your review, we have also provided a sample-level summary table of these 19 clinically diagnosed EPTB specimens and their corresponding test results here:

Sample Num.	Sample type.	Group	TB-DNA	Xpert	AFB	Culture	tNGS	IFN-gamma
113	Pus	Clinically diagnosed TB	Negative	Negative	Negative	Negative	Positive	Positive

131	Peritoneal fluid	Clinically diagnosed TB	Untested	Negative	Negative	Negative	Positive	Untested
21	Pleural fluid	Clinically diagnosed TB	Untested	Untested	Negative	Negative	Positive	Untested
25	Pus	Clinically diagnosed TB	Untested	Untested	Negative	Untested	Positive	Untested
31	Pus	Clinically diagnosed TB	Untested	Untested	Negative	Untested	Positive	Untested
88	Pus	Clinically diagnosed TB	Untested	Untested	Untested	Untested	Positive	Positive
85	Cerebrospinal fluid (CSF)	Clinically diagnosed TB	Untested	Negative	Negative	Negative	Negative	Positive
108	Pus	Clinically diagnosed TB	Untested	Negative	Negative	Negative	Negative	Positive
136	Pus	Clinically diagnosed TB	Untested	Negative	Negative	Negative	Negative	Positive
97	Pus	Clinically diagnosed TB	Negative	Untested	Negative	Negative	Negative	Positive
28	Cerebrospinal fluid (CSF)	Clinically diagnosed TB	Untested	Untested	Negative	Negative	Negative	Positive
168	Cerebrospinal fluid (CSF)	Clinically diagnosed TB	Untested	Untested	Negative	Untested	Negative	Positive
164	Cerebrospinal fluid (CSF)	Clinically diagnosed TB	Untested	Negative	Negative	Negative	Negative	Untested
95	Cerebrospinal fluid (CSF)	Clinically diagnosed TB	Untested	Untested	Negative	Negative	Negative	Untested
133	Cerebrospinal fluid (CSF)	Clinically diagnosed TB	Untested	Untested	Negative	Negative	Negative	Untested
149	Cerebrospinal fluid (CSF)	Clinically diagnosed TB	Untested	Untested	Negative	Negative	Negative	Untested
47	Pus	Clinically diagnosed TB	Negative	Negative	Negative	Untested	Negative	Untested
153	Cerebrospinal fluid (CSF)	Clinically diagnosed TB	Untested	Untested	Negative	Untested	Negative	Untested
20	Pus	Clinically diagnosed TB	Untested	Untested	Untested	Untested	Negative	Negative

16) Lines 216-218: The wording here is a bit weird. Please update for clarity.

Response:

Thank you for this comment. We have rephrased this sentence to improve clarity and parallel structure when comparing the three assays.

The revised text now reads (see **Lines 212–215**):

"In the PTB group, using the composite reference standard, tNGS achieved a sensitivity of 57.1%, a specificity of 97.1%, and an accuracy of 68.3%. In comparison, Xpert showed a sensitivity of 40.0%, a specificity of 100%, and an accuracy of 57.1%, while TB-DNA had a sensitivity of 46.3%, a specificity of 100%, and an accuracy of 60.0%."

17) Lines 223-224: How were the 30 samples chosen? Was this randomized in some way? Please clarify.

Response:

Thank you for this question. The 30 samples were not selected by randomization. For the head-to-head comparison between tNGS and Xpert, we retrospectively included all TB-positive specimens that (a) were positive by either Xpert or tNGS and (b) had valid results from both assays; no additional selection criteria were applied. Thus, this 30-specimen set represents the full concordance subset rather than a randomized sample.

As described in our response to Major Comment 2, we have revised the Results section to make this selection process explicit and to clearly

distinguish this 30-specimen concordance analysis from the resistance spectrum analysis in the full cohort (see **Lines 219–220**).

18) Line 229: "Nine" instead of 9.

Response:

Thank you for pointing this out. We have replaced "9" with the written form "Nine" at this position (see **Line 228**).

19) Line 230: "Three" instead of 3.

Response:

Thank you for this comment. We have replaced "3" with the written form "Three" at this position (see **Line 229**).

20) Line 241: EBV and CMV do not need to be italicized.

Response:

Thank you for pointing this out. We have removed italics from "EBV" and "CMV" at this position and ensured consistent formatting throughout the manuscript (see **Line 240**).

21) Line 245: Please update to "seven of the 13 samples (54%)" for consistency.

Response:

Thank you for this comment. We have updated the wording to “seven of the 13 samples (54%)” for consistency with the reported proportions (see **Line 244**).

22) Lines 246-247: This sentence is a little confusing as written. Is this saying that there was one case of PTB that had MTBC and NTM and one case of EPTB that had MTBC and NTM detected?

Response:

Thank you for pointing this out. Yes, the intended meaning was that one PTB sample and one EPTB sample showed co-detection of MTBC and NTM. To improve clarity, we have rephrased this part of the Results as follows (see **Lines 245–246**):

23) Line 284: Define DST.

Response:

Thank you for this comment. In line with our response to Major Comment 5, we have removed the sentence containing “DST” from the Discussion section.

24) All figures/tables: Define all acronyms used in the figure/table in the figure/table legend.

Response:

Thank you for this helpful suggestion. We have systematically reviewed all figures and tables and added abbreviation lists to each legend (see revised **Tables and Figures**) .

25) Supplemental methods (tNGS assay): Consider "custom" instead of "specially designed."

Response:

Thank you for this suggestion. We have replaced "specially designed" with "custom" in the description of the tNGS assay in the Supplemental Methods (see **Supplementary Methods, tNGS assay section**).

26) Supplemental Figure 1: Numbers don't match up in the pulmonary TB->clinically diagnosed TB (n=32 and then n=33).

Response:

Thank you for identifying this inconsistency. We have corrected the labeling in Supplemental Figure 1 so that the number of pulmonary clinically diagnosed TB cases is consistently shown as n = 32 throughout the figure (see **Supplemental Figure 1**) .

27) Supplemental Figure 1: So, NTM weren't detected in ANY non-TB cases? There were only two cases where an NTM was detected, and it

was detected with MTBC (See Minor Comment 22)? What is the purpose of showing the NTM arm in the figure? Only culture can detect NTM, so unless there is an NTM causing a false positive result, all of these should be negative.

Response:

Thank you for this insightful comment and for drawing attention to the interpretation of Supplemental Figure 1.

We would like to clarify that Supplemental Figure 1 only displays the detection of *M. tuberculosis* complex (MTBC) by different assays and does not summarize NTM detection results. In this figure, the “NTM” (or non-TB) branch refers to patients whose final clinical diagnosis was NTM disease or non-tuberculous disease, and it is included to show how many specimens in this clinically non-TB group were tested and how many were MTBC-negative by each method (i.e., to illustrate TB rule-out performance and the absence of false-positive MTBC results in this group). Thus, in this context, it is expected that the MTBC counts in the NTM/non-TB arm are zero, and the figure is not intended to represent NTM detection.

The actual detection of NTM organisms (including cases with MTBC–NTM co-detection) is presented in the main-text Figure 1A, where we explicitly mark all samples in which NTM were identified by culture or tNGS. As noted in our response to Minor Comment 22, there were two

samples in which tNGS co-detected MTBC and NTM—one in the PTB group and one in the EPTB group—and these are counted within the TB groups, not within the NTM/non-TB arm of Supplemental Figure 1. To avoid confusion, we have revised the legend of Supplemental Figure 1 to explicitly state that the figure summarizes MTBC detection only, and that NTM detection results are shown separately in Figure 1A (see Supplemental Figure 1 legend in the revised version). We retain the NTM/non-TB arm in Supplemental Figure 1 because it helps (a) delineate the full study population flow and (b) demonstrate that MTBC was not detected by any method in patients whose final diagnosis was non-TB/NTM disease, thereby complementing the TB rule-out aspect of the analysis.

28) Supplemental Table 5: Total=21 (not 20).

Response:

Thank you for catching this error. We have corrected the total in Supplemental Table 5 from 20 to 21 and carefully rechecked the entries to ensure internal consistency. We apologize for this oversight.

29) Supplemental Table 6: Please add that HHV-4 is EBV and that HHV-5 is CMV because those are how they are listed in the manuscript.

Response:

Thank you for this comment. In Supplemental Table 6, we have corrected the entries originally labeled as “Human herpes simplex virus 4” and “Human herpes simplex virus 5” to the appropriate names and aligned them with the nomenclature used in the main text (see **Supplemental Table 6**).

30) Supplemental Table 6: *Pneumocystis jirovecii* is misspelled.

Response:

Thank you for pointing this out. We have corrected the spelling of *Pneumocystis jirovecii* in **Supplemental Table 6**.

Reviewer #2 (Comments for the Author):

- There are several redundancies within line 155-156 could be made more succinct and clear

Response:

Thank you for this helpful comment. We have streamlined the description of patient enrollment and testing to remove redundancies and improve clarity, while retaining the key methodological information. (see **Lines 157–161**).

- The results section especially the first three paragraphs include a lot of

methodology that can be re-written to include within methods such that the results section can solely focus on conclusions and start off with Baseline clinical characteristics.

Response:

Thank you for this helpful suggestion. In the revised manuscript, we have streamlined the beginning of the Results section by removing methodological descriptions that were redundant with the Methods and focusing the opening on cohort composition and baseline characteristics.

Specifically, the original first two paragraphs have been condensed into a single, more concise paragraph (see **Lines 157–161**).

- It would significantly help the results section and the reader come to the same conclusions, if the author explicitly clarified which two assays are being compared and which one performed significantly better for eg. tNGS vs Xpert.

Response:

Thank you for this helpful suggestion. We have revised several parts of the Results section to make the comparisons between assays more explicit. In the paragraphs describing BALF, sputum, and EPTB samples, as well as in the overall sensitivity summary, we now clearly specify which assays are being compared (see **Lines 188, 194, 215**).

Re: Spectrum01698-25R2 (**Targeted Next-Generation Sequencing for Comprehensive Diagnosis and Drug Resistance Detection in Pulmonary and Extrapulmonary Tuberculosis: A Single-Center Retrospective Study**)

Dear Dr. Guiming Liu:

Happy New Year and thank you for re-submitting your work to Microbiology Spectrum. I am delighted to share that your manuscript has been accepted, and I am forwarding it to the ASM production staff for publication. Your paper will first be checked to make sure all elements meet the technical requirements. ASM staff will contact you if anything needs to be revised before copyediting and production can begin. Otherwise, you will be notified when your proofs are ready to be viewed.

Sincerely,
Kenneth Gavina
Editor
Microbiology Spectrum